

# Improved detection of global NOx emissions from shipping in Sentinel-5P TROPOMI data

Miriam Latsch[1], Andreas Richter[1], John P. Burrows[1], and Hartmut Bösch[1]

[1]Institute of Environmental Physics, University of Bremen (IUP-UB), Otto-Hahn-Allee 1, 28359 Bremen, Germany

**Correspondence:** Miriam Latsch (mlatsch@iup.physik.uni-bremen.de)

**Abstract.** Shipping is an important source of nitrogen oxide (NOx) emissions worldwide, contributing to air pollution and negatively affecting marine environments, ecosystems, and biodiversity. TROPOMI (TROPOspheric Monitoring Instrument) aboard the Sentinel-5 Precursor (S5P) has significantly enhanced the ability to detect ship emissions from space due to its low measurement noise levels and high spatial resolution of 5.5 x 3.5 $km^2$. This study uses the TROPOMI tropospheric $NO_2$ slant
column density (tSCD) to identify global shipping routes. Preprocessing techniques, including iterative high-pass and Fourier filtering, markedly improve the detection of shipping lanes, revealing previously undetectable routes. Our analysis examines the impact of high-pass filter box sizes, demonstrating that smaller sizes enhance the visibility of narrow shipping features, while larger box sizes increase overall $NO_2$ signals. Additionally, we investigate various flagging criteria that affect $NO_2$ signal distribution, highlighting the critical importance of careful selection for accurate emission monitoring. Filtered TROPOMI $NO_2$
tSCDs over oceans show a strong correlation with shipping activities, as confirmed by comparison with the CAMS-GLOB-SHIP (Copernicus Atmospheric Monitoring Service for Global Shipping) inventory, and also reveal unknown shipping routes in regions such as the Bering Sea. Furthermore, TROPOMI effectively captures $NO_2$ emissions from offshore oil and gas platforms, with $NO_2$ hotspots in the TROPOMI data aligning well with locations of offshore installations listed in the OSPAR (Oslo and Paris Commission) and BOEM (Bureau of Ocean Energy Management) inventories. Lastly, the filtered TROPOMI
$NO_2$ tropospheric vertical column densities (tVCDs) are compared with the tVCDs from the CAMS (Copernicus Atmospheric Monitoring Service) model, which has a coarse spatial resolution of 0.4°. While both data sets effectively identify global shipping lanes, the CAMS $NO_2$ tVCDs are significantly higher compared to the filtered TROPOMI tVCDs, with differences of up to a factor of 100 in the South Atlantic Ocean.

## 1 Introduction

Nitrogen oxide (NO) and nitrogen dioxide ($NO_2$) are chemically coupled in the troposphere, and their sum is referred to as NOx, where $NOx = NO + NO_2$. They play a key role in the formation of tropospheric ozone, lead to acid rain, contribute to aerosol formation, and can increase nutrient input into water bodies. NOx is primarily released from all combustion processes, bacterial soil emissions, and lightning. The combustion of fossil fuels in the transportation sector, in particular, is a major tropospheric source of NOx emissions. Global shipping is the cornerstone of international trade, accounting for approximately
90% of worldwide goods transportation, as reported by the International Maritime Organization (IMO). Between 2012 and



2018, the greenhouse gas emissions of total shipping increased by 9.6% (Faber et al., 2020). The rise in ship emissions is particularly concerning in port cities and coastal areas, where NOx significantly impacts air quality and human health. In 2019, the International Council on Clean Transportation (ICCT) reported that air pollution from the about 70,000 international ships operating globally contributes approximately 15% to early mortality (Anenberg et al., 2019). Given the substantial impact
of shipping to NOx emissions, it is critical to understand and regulate these emissions to mitigate their harmful effects. To address this, the IMO has defined three limits for NOx emissions from ships (Tier 1-3) in the International Convention for the Prevention of Pollution from Ships (MARPOL) ANNEX VI, based on the engine power and the ship's construction date. Additionally, stringent international emission regulations have been established in designated Emission Control Areas (ECAs) to reduce NOx emissions from shipping, such as in the North American waters since 2016 (U.S. EPA, 2010) and in the North
and Baltic Seas since 2021 (IMO, 2023).

The retrieval of tropospheric $NO_2$ slant and vertical columns began with the inversion of the nadir sounding observations of the upwelling solar visible radiation at the top of the atmosphere by the GOME instrument on ESA ERS-2 (Burrows et al., 1999). Satellite instruments have previously been used to track NOx emissions from ships along some of the busiest shipping lanes from space. The measurements used for this application have been made by the following instruments: GOME,
SCIAMACHY, OMI, and GOME-2. These studies have provided crucial insights into the spatial and temporal distribution of ship emissions and their effects on air quality and climate. For instance, the shipping lane from India to Indonesia was first visualized using GOME data by Beirle et al. (2004). SCIAMACHY measurements have been used to detect the shipping routes in the Red Sea, the Persian Gulf, and toward China and Japan (Richter et al., 2004; Franke et al., 2009). OMI data has identified the shipping lanes in the Mediterranean Sea, around the Iberian Peninsula, and in the Baltic Sea towards the English Channel
(Marmer et al., 2009; Ialongo et al., 2014; Vinken et al., 2014). Similarly, GOME-2 data has detected the shipping lanes around the African continent (Richter et al., 2011).

Compared to earlier satellite instruments, TROPOMI (TROPOspheric Monitoring Instrument) onboard the Sentinel-5 Precursor (S5P) offers improved capabilities for estimating ship emissions due to the high signal-to-noise ratio in the $NO_2$ measurements (Veefkind et al., 2012). Additionally, its high spatial resolution of 5.5 x 3.5 $km^2$ enables relatively small-scale but
significant emissions, such as plumes from individual ships, to be retrieved in sun glint geometry, as demonstrated by Georgoulias et al. (2020) in the Mediterranean Sea. Riess et al. (2022) investigated the ship emissions reduction in European seas due to decreased economic activities during the COVID-19 pandemic. Furthermore, artificial intelligence methods have been used to identify single shipping plumes in the Mediterranean Sea (Kurchaba et al., 2022). These findings underscore the potential of TROPOMI data to improve our understanding of ship emissions and their environmental impacts.

The method presented in this research aims to improve the detection of global shipping-related NOx emissions, also referred to as shipping signals, in filtered mean S5P TROPOMI $NO_2$ data. Beirle et al. (2004) has already presented a high-pass filter method to estimate $NO_2$ vertical columns in GOME measurements. Latsch et al. (2023) applied a high-pass filtering method to TROPOMI $NO_2$ measurements for the first time to facilitate the identification of ship emissions. Using a similar approach, Pseftogkas et al. (2024) has demonstrated that changes in maritime trade routes, such as those caused by the Red Sea shipping



crises, can be tracked. Here, this method is further improved in order to reduce the detection limit and identify as many shipping
signals in TROPOMI data as possible.

In Sect. 2 of this manuscript, the data selection and methodology are presented in detail. Sect. 3 presents the results of
detecting global ship emissions in the TROPOMI NO$_2$ data with our filtering method. Sect. 3.1 focuses on the influence of
different flagging criteria on the data and a separated shipping lane. In Sect. 3.2, our method is applied to identify offshore oil

and gas platforms. In Sect. 4, the TROPOMI measurements are compared with CAMS model data. Finally, Sect. 5 summarizes
the findings of this study.

## 2  S5P TROPOMI data processing

In this study, the S5P TROPOMI NO$_2$ RPRO and OFFL Level 2 data of processor versions 2.4.0 to 2.6.0 are used (van Geffen
et al., 2022; Eskes et al., 2023). Specifically, this work focuses on the tropospheric NO$_2$ slant column density (tSCD). However,

this raises the question of why we focus on the tSCD as a critical parameter for detecting shipping signals. In the differential
optical absorption spectroscopy (DOAS) method, the air mass factor (AMF) is typically used to convert the slant column
density (SCD) into the vertical column density (VCD) using the relation $VCD = \frac{SCD}{AMF}$. The AMF is usually computed with a
radiative transfer model, such as SCIATRAN (Rozanov et al., 2005), and accounts for the processes and effects that influence
the radiative transfer and the resulting path of electromagnetic radiation through the atmosphere between the sun and the

satellite instrument. These factors include the following: the viewing and solar geometry, the fitting window, the surface albedo
and pressure, the NO$_2$ vertical profile, and the presence of clouds and aerosols. Consequently, the tropospheric VCD (tVCD) is
the geophysically meaningful quantity to be used in the quantitative analysis of TROPOMI observations. The primary reason
for using the tSCD instead of the tVCD lies in how the AMF accounts for the vertical distribution of NO$_2$ in the atmosphere.
The AMF uses vertical profiles from atmospheric models, which incorporate information on global shipping routes and predict

NO$_2$ to be concentrated lower in the troposphere where ships emit their exhaust plumes. Typically, NO$_2$ in the troposphere
over the open sea is relatively uniformly distributed with height, resulting in a comparatively large AMF. However, along
shipping routes, most of the NO$_2$ in the troposphere is located near the sea water surface, leading to smaller AMF values.
This fact explains why shipping routes are visible in spatial AMF maps. When these AMFs are applied to maps with constant
tSCDs, the tVCDs over shipping lanes known to the model appear higher, even if the measured tSCDs do not directly capture

shipping signals. Regarding this, Riess et al. (2023) investigated AMF values based on aircraft NO$_2$ profiles, showing high NO$_2$
values near the water surface over the North Sea. In addition, the AMF in the operational TROPOMI product is derived using
NO$_2$ profiles from the global atmospheric chemistry transport model TM5 (Tracer Model, version 5), which has a relatively
coarse spatial resolution. Consequently, the AMF values are limited in accurately representing the fine-scale variations in NO$_2$
concentrations along the shipping lanes. In contrast, the tSCD is independent of model information about the location and

intensity of shipping routes and offers a finer spatial resolution than the TM5. Therefore, we deduce that the NOx emissions
from ships are most accurately determined from an analysis of the tSCD. In addition, our study applies the TROPOMI surface
classification mask to examine only pixels over water to focus specifically on ship emissions.



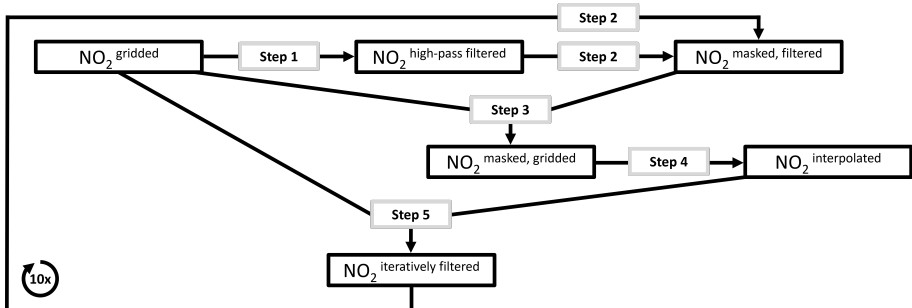

**Figure 1.** Schematic overview of the iterative high-pass filtering procedure applied on the gridded TROPOMI NO$_2$ tSCD. After a thorough examination, an iteration number of ten times proves to be the most efficient.

During data processing, distinct box-like patterns were noticed in the TROPOMI NO$_2$ data. These artifacts result from the stratospheric correction applied to the operational TROPOMI data, which integrates the stratospheric model columns over all altitude levels above the tropopause height. The problem arises because the tropopause height is obtained from a data set with a spatial resolution of 1° that is not interpolated. As a result, block-like patterns appear in both the stratospheric and tropospheric columns along the borders of the tropopause pixels. Although the NO$_2$ changes between these boxes are relatively small, they significantly influence the following data analysis. Spatial averaging is applied to the stratospheric VCD (sVCD) to address this issue, resulting in the smoothed sVCD. This preprocessing step effectively reduces the presence of the box-like artifacts and improves the accuracy of the further analysis. After smoothing the sVCD, the tSCD is recalculated using Eq. (1), where $totalSCD$ is the total SCD from the DOAS fit, $destrSCD$ is the across-track NO$_2$ slant column stripe offset provided in the TROPOMI NO$_2$ product, labeled as "nitrogendioxide_slant_column_density_stripe_amplitude", $sVCD^*$ is the smoothed sVCD, and $sAMF$ is the stratospheric AMF which is also taken from the TROPOMI product.

$$tSCD = totalSCD - destrSCD - sVCD^* \cdot sAMF \tag{1}$$

The resulting tSCDs are gridded to a resolution of 0.03° x 0.03° and averaged from 1 May 2018 to 30 April 2024, in total six years. On this average, a high-pass filter with a standard box size of approximately 1° in both longitude and latitude is applied to extract the shipping lanes (see Appendix A1 for more details on the high-pass filtering method). This approach assumes that shipping signals are present as localized enhancements of NO$_2$ over ocean regions. However, due to taking the rolling mean on the data, the high-pass filtered tSCDs show a NO$_2$ saturation at coastlines and large negative values around the highlighted shipping lanes. To reduce these artifacts, the NO$_2$ tSCDs are iteratively high-pass filtered, as shown in Fig. 1, including the following procedure steps:

- Step 1: The high-pass filtering is applied on the gridded NO$_2$ tSCD values (NO$_2^{\text{gridded}}$), as described above and in Appendix A1 in detail, resulting in the high-pass filtered NO$_2$ tSCDs (NO$_2^{\text{high-pass filtered}}$).

- Step 2: A threshold value of $\pm$3e13 molec cm$^{-2}$ is defined on the NO$_2^{\text{high-pass filtered}}$ tSCDs to eliminate values larger or smaller than this threshold, resulting in the masked, high-pass filtered NO$_2$ tSCDs (NO$_2^{\text{masked, filtered}}$).





- – Step 3: The mask defined by the last step is applied on the $NO_2^{gridded}$ tSCDs, resulting in the masked, gridded $NO_2$ tSCDs ($NO_2^{masked, gridded}$).

- – Step 4: The $NO_2^{masked, gridded}$ tSCDs are calculated by linear interpolation, resulting in the interpolated $NO_2$ tSCDs ($NO_2^{interpolated}$).

120 – Step 5: The $NO_2^{interpolated}$ tSCDs are used to high-pass filtering the $NO_2^{gridded}$ tSCDs by subtracting the rolling mean with the same standard box size, resulting in the iteratively filtered $NO_2$ tSCDs ($NO_2^{iteratively filtered}$).

This procedure is repeated ten times such that after the first iteration, the $NO_2^{iteratively filtered}$ tSCDs are used to generate the mask in Step 2. The threshold value of $\pm 3e13$ molec cm$^{-2}$ was chosen in order to be able to detect the smallest possible signals, but at the same time not to fall below the detection limit of the averaged $NO_2$ data. In addition, the threshold value 125 was selected for both positive and negative values to avoid a high bias in the filtered $NO_2$ data. The magnitude of the ship emission contribution to the tSCD may be more accurate using this iterative approach, among other things, by reducing the $NO_2$ saturation near most of the land masses and raising the non-physical negative $NO_2$ values around the shipping lanes. After applying the high-pass filter, stripe-like patterns oriented in the direction of the TROPOMI orbits emerge due to the limited number of varying overpass patterns in the S5P orbit. Because these patterns limit the detection of shipping signals, a Fourier 130 filtering method is applied to mitigate the impact of the orbital structure in the background (see Appendix A2 for more details on the Fourier filtering method). These filtering approaches enhance the identification of shipping routes in the TROPOMI $NO_2$ data.

Several data selection criteria can be utilized to flag the filtered TROPOMI $NO_2$ data for specific meteorological and measurement geometry conditions to evaluate the impact of observational conditions on the analysis. This approach enables the 135 analysis of which flagging variables impact the detection of the shipping signals and how they do so. The quality (qa) flagging includes only pixels with a quality value larger than 0.75, excluding measurements containing a cloud radiance fraction exceeding 50%, having a large RMS, or exhibiting a very small AMF. The cloud fraction (CF) flagging considers only pixels with a cloud cover less than 0.5 to eliminate measurements affected by an extensive cloud cover. The cloud height (CH) flagging uses only pixels without clouds or with clouds at altitudes below 2 km to avoid higher clouds that may shield the shipping 140 $NO_2$ from the satellite view. The wind speed (wind) flagging retains only pixels with wind speeds between 0 and 5 m s$^{-1}$ to exclude conditions where the $NO_2$ is rapidly dispersed due to strong advection and related mixing. Lastly, the sun glint (sg) flagging identifies pixels defined as "sun glint is possible" (indicated by the TROPOMI variable "geolocation_flag = 2"), enabling the investigation of situations where the sensitivity for $NO_2$ near the sea surface is potentially increased due to the direct reflection of sunlight off ocean surface waves towards the satellite. The effect of these different data flagging criteria 145 depends on the specific context in which they are applied. Thus, sun glint conditions can increase TROPOMI's sensitivity to detect $NO_2$ emissions from shipping, as previously demonstrated by Georgoulias et al. (2020) for individual ship plumes and Riess et al. (2022). However, one issue in applying these flagging criteria is that the background noise increases due to the reduced number of available measurement values.





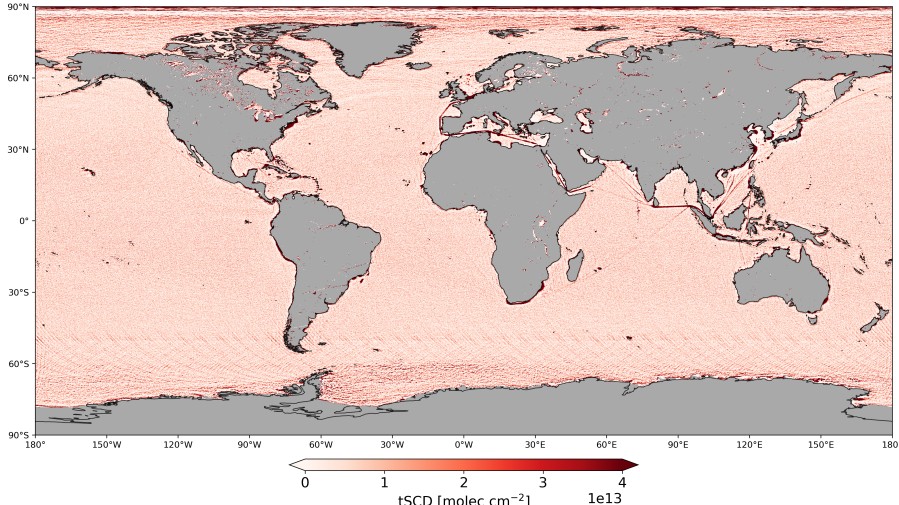

**Figure 2.** Overview of the NO$_2$ signals from six years of filtered TROPOMI data. Many shipping routes are identified, such as known ones, e.g., in the Mediterranean Sea, the Red Sea, and the Indian Ocean, and ones that have not been detected before in satellite data, e.g., towards the Panama Canal, between Indonesia and Australia, and between Asia and North America. A high-resolution, zoom-in version of this global map is available at: https://www.iup.uni-bremen.de/doas/tropomi_ships.htm.

## 3 Global ship emissions

This section presents several regions where shipping signals in the filtered TROPOMI NO$_2$ data are detected using the introduced method. First, Fig. 2 shows a global map providing an overview of the detected NO$_2$ signals over the six years of analysis (from May 2018 to April 2024). As the standard criterion, no data flagging is applied to the filtered TROPOMI NO$_2$ tSCDs, as discussed later. A large number of shipping routes are directly visible on the global map, both the known shipping lanes, e.g., in the Mediterranean Sea, the Red Sea, and the Indian Ocean, and the previously undetected shipping lanes, e.g., in

the Caribbean Sea to the Panama Canal, between Indonesia and Australia, and between Asia and North America. All regions where NO$_2$ signals from shipping are identified will be presented later in more detail. It is worth noting that all these shipping routes come directly from the TROPOMI NO$_2$ data without any model a priori.

Even though many shipping routes can be detected in the filtered TROPOMI NO$_2$ data, the global map overlaps with background noise. The noise results from using the high-pass filter, highlighting the very small NO$_2$ signals from ships but

simultaneously amplifying the background noise. The stripe-like patterns oriented in the TROPOMI orbit direction are probably artifacts caused by the periodic overlapping of the S5P orbits. The Fourier filtering could not altogether remove them. In addition, the systematic artifacts around some land masses warrant careful consideration. These elevated NO$_2$ columns, particularly along coastlines and near islands, result from DOAS retrieval fit problems in inhomogeneous scenes where large intensity differences between adjacent measurements prevail, for example, along the sea ice edge or around permanently cloudy

regions (Richter et al., 2020). This scene inhomogeneity effect is further amplified by the high-pass filtering, affecting the ac-



curacy of TROPOMI $NO_2$ measurements. This phenomenon and AMF effects are visible in the polar regions, where a distinct contrast appears between the typical background and a patterned distribution of $NO_2$ signals. Off the coast of Antarctica and the Arctic, these pronounced structures represent the average sea ice extent. The variation in $NO_2$ signals along the borders of ice and land surfaces is primarily caused by differences in the surface albedo, specifically, the fraction of sunlight reflected by a surface. Sea ice has a high albedo, reflecting a significant portion of sunlight, while the open ocean, with its lower albedo, absorbs most of the incoming light. This sharp contrast in reflectivity between the bright sea ice and the darker ocean surface results in a stronger $NO_2$ signal over sea ice, even when the actual $NO_2$ concentrations remain unchanged due to AMF effects. This AMF effect can also be observed over bright clouds, where the increased reflectivity also enhances the $NO_2$ signal in the measurements (Latsch et al., 2022). Applying the iterative filtering approach to the TROPOMI $NO_2$ data reduces the presence of these phenomena.

However, not all elevated $NO_2$ signals in polar regions are artifacts. For example, higher $NO_2$ signals are detected in the Bering Sea, which will be discussed later in Fig. 5, and in the Ross Sea near the Antarctic continent (Appendix Fig. B1a). As the CAMS-GLOB-SHIP data indicates (Appendix Fig. B1b), the elevated $NO_2$ signal in the Ross Sea can be assigned to shipping activities. The data reveal shipping routes to and from various research stations near the coast, including the Jang Bogo station, Gondwana station, Mario Zucchelli station, and the McMurdo station on Ross Island. The filtered $NO_2$ TROPOMI data were examined for the two polar seasons to ensure that the elevated signal may be attributed to shipping and not only to the research stations (not shown). Only in the summer season with minimal sea ice extent (June to November) the higher $NO_2$ signal is detected, while during the winter season (December to April), the characteristic sea ice pattern dominates the TROPOMI $NO_2$ data in the polar region.

Figure 3 shows the seas surrounding the European continent, using different box sizes for both the longitude and latitude in the rolling mean applied to the high-pass filter. The choice of the box size defines the spatial size of $NO_2$ enhancements that are detected and, therefore, influences the characteristics and number of visible shipping routes. The primary route in the Mediterranean Sea, extending from the exit of the Suez Canal to the Strait of Gibraltar and continuing across the North Atlantic Ocean towards the English Channel, is consistently detected regardless of the box size. In contrast, more minor shipping routes, such as the direct shipping lane in the Black Sea leading into the Aegean Sea, encircling the Greek islands before reaching the Ionian Sea, and those in the Adriatic and Tyrrhenian Seas around Italy, particularly between Sardinia and Corsica, are only faintly visible when using a smaller box size of approximately 0.25° (Fig. 3a). These routes grow increasingly pronounced with a larger box size of about 1° (Fig. 3b). Additionally, weaker signals in the North Sea and Baltic Sea are more visible with a 1° box size. However, the visualization of separated shipping lanes, such as the two one-directional shipping routes in the North Atlantic Ocean, which here is for the first time demonstrated in $NO_2$ satellite data, is only possible with a smaller box size (see also Sect. 3.1 for a detailed analysis of this lane separation). The $NO_2$ saturation near the coasts in the North Sea, around Italy, and of France, Spain and North Africa, especially visible in Fig. 3b, is a result of the iterative high-pass filtering which partially enhances the higher $NO_2$ signals near land with the same efficiency as the shipping lanes. Overall, a box size of 1° appears to be the most effective in highlighting the global shipping signals. Consequently, this study adopts a 1° box size as the standard, balancing detail with clarity to visualize shipping routes, especially on a global scale.




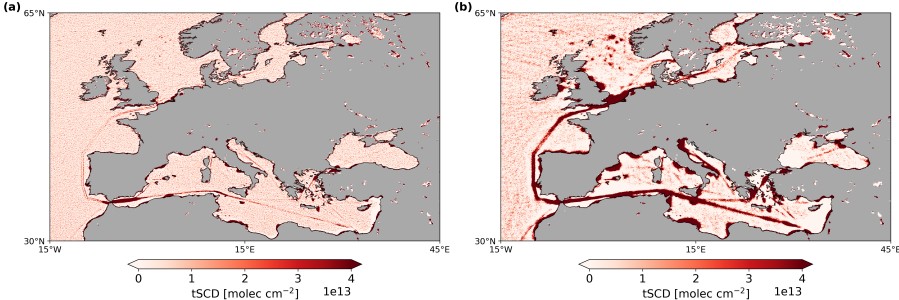

**Figure 3.** Filtered TROPOMI NO$_2$ signals from shipping activity in the European seas, analyzed with high-pass filters at different box sizes: approximately (a) 0.25° and (b) 1° in both longitude and latitude. The 0.25° box size reveals details of more minor shipping routes, such as the two shipping lanes in the North Atlantic Ocean, while weaker signals, such as in the Baltic Sea, and NOx emissions from oil and gas platforms in the North Sea (see also Fig. 9), are more visible with the 1° box size. The NO$_2$ saturation near land, especially visible in the North Sea, around Italy, and at the coast of France, Spain and North Africa in (b), is a result of the iterative high-pass filtering which partially enhances the higher NO$_2$ signals at coastlines.

Previous studies have demonstrated that the prominent shipping lane in the Red Sea is detectable by various satellite instruments, including TROPOMI. Figure 4a shows high NO$_2$ values in this region, effectively highlighting the shipping lane as it enters the Arabian Sea. There, the route diverges in multiple directions, southward toward the Laccadive Sea and northward along the coast of Oman into the Gulf of Oman. No clear NO$_2$ signal is visible in the Persian Gulf because the limited spatial

extent of the Gulf and its proximity to the landmasses lower the ability to detect NO$_2$ emissions using the iterative high-pass filtering method with a 1° box size. Only the non-iteratively filtered NO$_2$ TROPOMI data with a smaller box size of 0.25° proves to be highly effective in visualizing the shipping route near the coast of Iran in this spatially limited region (see Appendix Fig. B2). Additionally, the NOx emissions from offshore platforms are distinctly visible as isolated points in the middle of the Persian Gulf with the 0.25° box size. Furthermore, two separated shipping lanes in the Gulf of Aden, leading to and from

the Red Sea, are also identified. However, it is important to note that the non-iteratively filtered NO$_2$ data with the 0.25° box size does not capture signals in the Arabian Sea as visibly as in the iteratively filtered NO$_2$ data with the 1° box size.

Figure 4b presents a region where shipping signals have not been previously detected. In the Gulf of Mexico, two prominent shipping lanes originate near the port of Houston, intersecting again in the middle of the Gulf. One of these routes extends as far as the North Atlantic Ocean near the Bahamas, while the other one continues into the Caribbean Sea. In this region, a distinct

NO$_2$ signal traces a direct path from the strait between Cuba and Haiti to the Panama Canal, where the NO$_2$ concentrations are markedly elevated. Some outflow of NO$_2$ from the Caribbean Islands is also visible but probably not linked to shipping activities and enhanced at some coastlines due to the iterative high-pass filtering. However, NOx emissions from offshore oil and gas platforms are identified in the Gulf of Mexico (see Sect. 3.2 for more details). In the Pacific Ocean, a single shipping signal is detected, stretching from the Panama Canal along the Mexican coast up to Northern America. Notably, higher NO$_2$



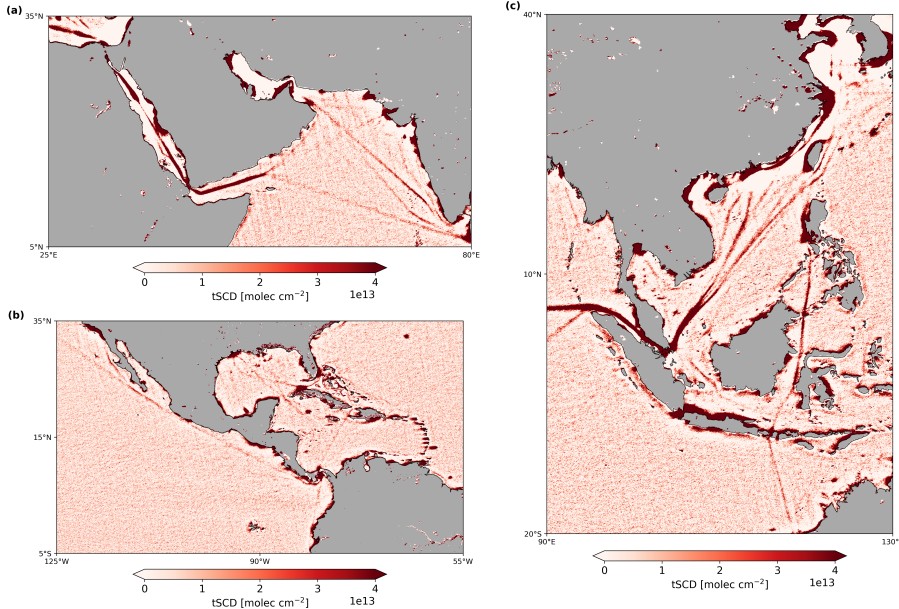

**Figure 4.** Filtered TROPOMI NO$_2$ signals from shipping activity in (a) the Red and Arabian Seas, (b) the Seas around Middle America, and (c) the South China Sea. NOx emissions from offshore platforms are visible as elevated dot-shaped NO$_2$ values, e.g., in (a) the Persian Gulf, (b) the Gulf of Mexico, and (c) the Gulf of Thailand (see also Sect. 3.2).

values are visible along the coast of the Baja California peninsula. Further south, a shipping route extends from the Panama Canal towards Ecuador and into the southern Pacific.

In Fig. 4c, a prominent shipping lane is visible in the Andaman Sea, where it is joined by a subtler route originating from South Africa (see Fig. 2), continuing through the Strait of Malacca toward Singapore. Farther northwest, this shipping lane divides into several distinct lines in the South China Sea, the first time these shipping routes have been identified in TROPOMI

NO$_2$ data. Additionally, a clear NO$_2$ signal is detected along the coast of China, stretching from the South China Sea across the East China Sea and into the Yellow Sea. Two separated lanes are also visible in the Philippine Sea, just off the coast of Taiwan. Another clear shipping route extends directly from the Philippines, passing through the Makassar Strait and towards Australia. In the Gulf of Thailand, only a faint shipping signal is detected, which can be attributed to the 1° box size used in the high-pass filtering, as it is too coarse to capture fine details in this spatially limited region. However, elevated NO$_2$ values

are visible as pronounced dots corresponding to NOx emissions from offshore platforms. Along the southern coast of many islands, a region of enhanced NO$_2$ is found, on the one hand, another example of the artifact created by inhomogeneous scenes at some coastlines, and on the other hand, a result of the iterative filtering method.

A distinct shipping route from the Korea Strait to the Tsugaru Strait in the Sea of Japan is detected in the filtered TROPOMI NO$_2$ data, as shown in Fig. 5a. Extending from the Tsugaru Strait, two shipping lanes are identified for the first time in satellite

data in the North Pacific. The more southerly NO$_2$ signal, which connects to the shipping lane originating from the Panama





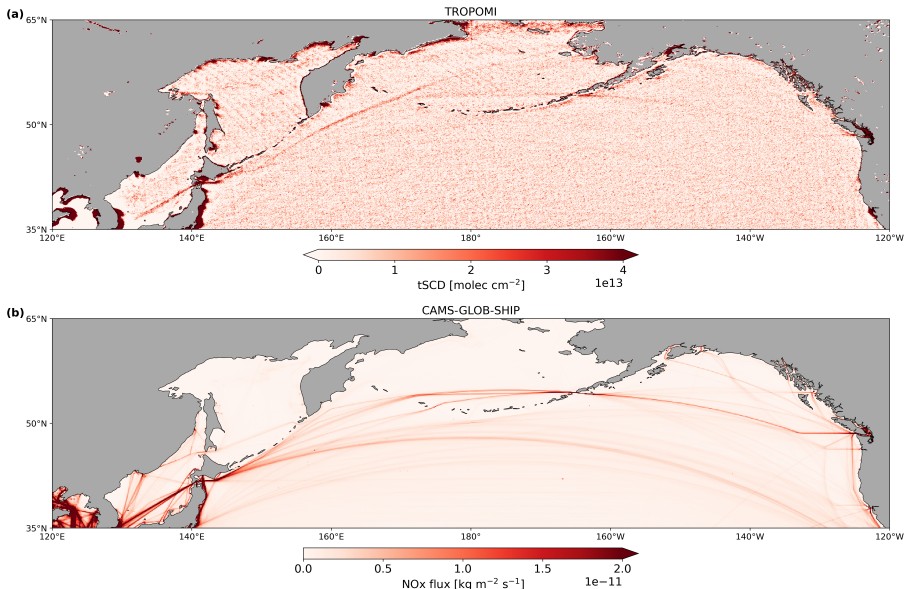

**Figure 5.** (a) Filtered TROPOMI and (b) CAMS-GLOB-SHIP NO$_2$ signals from shipping activity in the North Pacific correlate well for the two shipping routes between Asia and North America. The NO$_2$ signal in the Bering Sea is only identified in the filtered TROPOMI data and not tracked through the AIS-based CAMS-GLOB-SHIP data, possibly due to disabled trackers onboard military or unauthorized fishing vessels. The CAMS-GLOB-SHIP data cover the years from 2018 to 2021.

Canal, is noticeably weaker than the northern lane, which diverges into two branches in the Bering Sea. One branch extends across the southern Bering Sea, reaching the Aleutian Islands (Alaska, USA) and continuing along the Salish Sea south of Vancouver Island (Canada). The other branch appears to go to the Alaskan mainland just north of the Nunivak Island. This northern signal, previously undetected in satellite data and not typically associated with known shipping routes, indicates a
need for comparison with the CAMS-GLOB-SHIP (Copernicus Atmospheric Monitoring Service for Global Shipping) data. CAMS-GLOB-SHIP is a global emission inventory based on AIS (Automatic Identification System) information and the Ship Traffic Emission Assessment Model (STEAM) to describe ship traffic activity. The CAMS-GLOB-SHIP inventory, version 3.2, provides global ship emissions on a 0.1° x 0.1° grid for the years 2018 to 2021, available from the ECCAD (Emissions of Atmospheric Compounds and Compilation of Ancillary Data) database (ECCAD, 2018; Granier et al., 2019). Notably, the
CAMS-GLOB-SHIP data show no significant ship traffic activity in the northern Bering Sea, where this elevated NO$_2$ signal is detected in the TROPOMI data (Fig. 5b). This discrepancy suggests that ship activities in this region may not be fully captured in the CAMS-GLOB-SHIP inventory due to disabled AIS trackers, possibly linked to illegal or military operations. It is not uncommon for some ships, particularly military or unauthorized vessels, to turn off their AIS trackers, preventing the detection of their presence in certain areas, including in the CAMS-GLOB-SHIP data. Supporting this, the study by Welch et al. (2022)
notes that the fraction of estimated total fishing vessel activity obscured by suspected AIS disabling events is highest in the Northwest Pacific. This finding corresponds to the elevated NO$_2$ signal detected in the TROPOMI data for this region in the





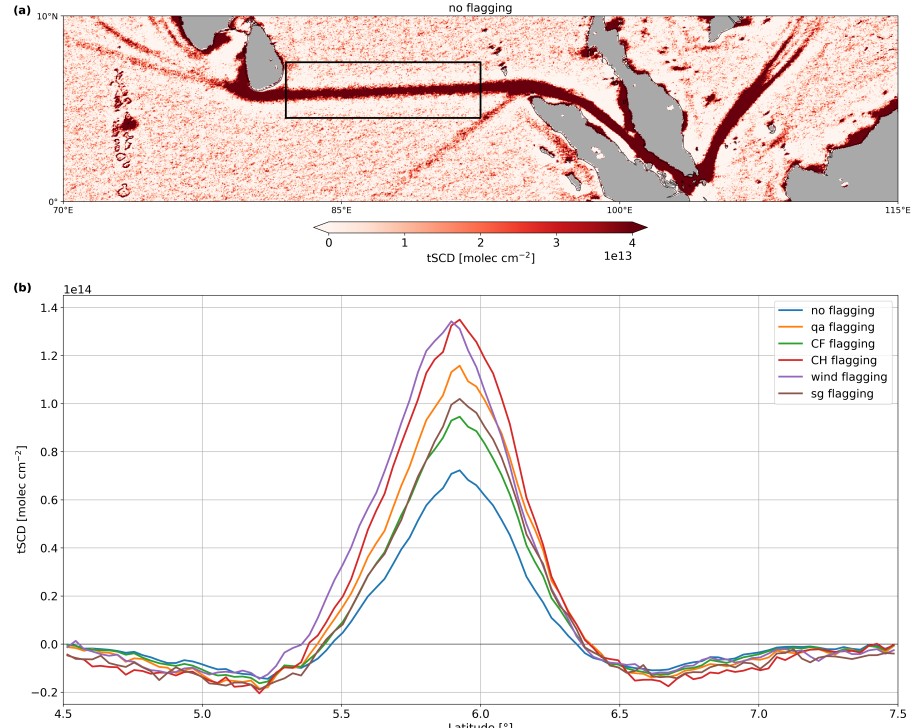

**Figure 6.** (a) Filtered TROPOMI NO$_2$ signals from shipping activity in the Indian Ocean, between Sri Lanka, Indonesia, Malaysia, and Singapore, without data flagging. The black rectangle highlights the area shown in (b), which displays the cross-sections of the shipping lane for the different flagging criteria: all pixels included (blue), and only pixels including qa>0.75 (orange), CF<0.5 (green), CH<2 km (red), 0 m s$^{-1}$<wind speed<5 m s$^{-1}$ (purple), and sg possible situations (brown). The curves represent the longitudinal average from 82°E to 92.5°E as a function of latitude.

absence of AIS signals despite sufficient satellite reception quality. Therefore, this research supports the hypothesis that the signal is real but cannot be tracked through AIS data due to disabled trackers.

## 3.1 Shipping lane cross-sections

The distinct shipping route from Sri Lanka to the Strait of Malacca, located between Malaysia and Indonesia, has already been discussed in other publications and is also clearly visible in the filtered TROPOMI NO$_2$ data (Fig. 6a). Here, we focus on the cross-section of this shipping lane, marked by the black rectangle in Fig. 6a, to investigate the impact of various flagging criteria (see Sect. 2) on the NO$_2$ signal distribution along the shipping lane. Figure 6b displays the longitudinal average of the NO$_2$ tSCD from 82°E to 92.5°E, plotted as a function of latitude from 4.5°N to 7.5°N for the different flagging criteria. All 260 curves show a prominent peak at approximately 5.9°N and two minima at around 5.2°N and 6.7°N. It should be mentioned that the NO$_2$ distribution heights and widths depend on the number of iterations of the high-pass filtering procedure: the more iterations, the higher and broader the peaks, as less diluted NO$_2$ from ships is included in the smoothed background map.



As expected, the NO$_2$ column peak is lowest, and the background values are highest when no flagging is applied, resulting in the lowest variability among the curves. Applying flagging criteria leads to higher NO$_2$ peaks with differences of at least 2e13 molec cm$^{-2}$ for the curve flagged for cloud fraction (CF) and up to 6e13 molec cm$^{-2}$ for the curves flagged for cloud height (CH) and wind. By comparison, the differences at the minima or more general for the background values are lower, not exceeding 1e13 molec cm$^{-2}$. The negative values are attributed to the high-pass filter, which emphasizes the larger shipping signal values while causing the background values directly adjacent to the shipping lanes to appear negative due to the rolling mean applied in this method. The six curves show less variation as the distance from the peak of the shipping route increases, where the rolling mean has less influence than around the highest values. Using the iterative filtering method, the values around the shipping lanes are not as negative as when applying the high-pass filter only once.

Regarding the NO$_2$ signal peak, flagging for small CFs and sun glint (sg) has the lowest effect on the NO$_2$ signal distribution, aside from no flagging. When quality (qa) flagging is applied, the NO$_2$ peak increases slightly, although not as much as with the CH and wind flagging criteria. The latter is notably shifted for low wind speeds. This shift may be attributed to a correlation between wind speed and direction. Beirle et al. (2004) identified a strong seasonal asymmetry in the NO$_2$ signal peak due to the impact of the Intertropical Convergence Zone (ITCZ) on the ship track in this region. In calm wind conditions, the NO$_2$ emissions from ships may accumulate and form a more concentrated plume at a specific location. Conversely, in higher wind conditions, captured by the other flagging criteria, the NO$_2$ from ships disperses more quickly, leading to a more spread-out signal. Therefore, the peak for low wind speeds may be shifted toward the location of the shipping lane where the NO$_2$ accumulates. Flagging for the lowest CHs is among the most effective criteria due to the increased NO$_2$ peak. One potential reason for this higher peak is that low clouds may shield NO$_2$ and trap it closer to the surface, resulting in higher NO$_2$ concentrations within the shipping plumes. Additionally, low clouds can reflect more sunlight toward the Earth's surface, enhancing the sensitivity of the TROPOMI instrument for NO$_2$ above the clouds. This increased sensitivity may lead to higher measured concentrations of NO$_2$ in the shipping lanes when only the lowest CHs are considered. It should be noted that the flagging criteria have different effects on the detected NO$_2$. For example, wind flagging selects situations where there is actually more NO$_2$ in the atmosphere, while the other flagging criteria select situations that improve the visibility of NO$_2$. The latter can, at least in principle, be compensated by a suitable AMF, while the higher NO$_2$ values at low wind speeds are real and should also appear in the tVCDs, as shown by Riess et al. (2022).

Overall, the choice of flagging criteria significantly impacts the observed NO$_2$ signal along the shipping lanes. Based on the results discussed above, one would therefore apply qa flagging and possibly also flagging for low clouds to improve the detection of weak shipping signals. However, applying these criteria has challenges, primarily due to the reduced number of available measurements. This reduction leads to a deterioration in the signal-to-noise ratio, as shown in Fig. 7a-f, which presents the shipping lane in the eastern North Atlantic region under different flagging conditions. The background noise is low when no flagging is applied (Fig. 7a). In contrast, the background noise significantly increases with the application of the flagging criteria (Fig. 7b-f), particularly sg flagging (Fig. 7f), due to the strongly reduced number of measurements. Furthermore, a pronounced artifact emerges south of Great Britain, mirroring the shape of the sg flagging area in the TROPOMI data but bearing no relation to any actual shipping signals.




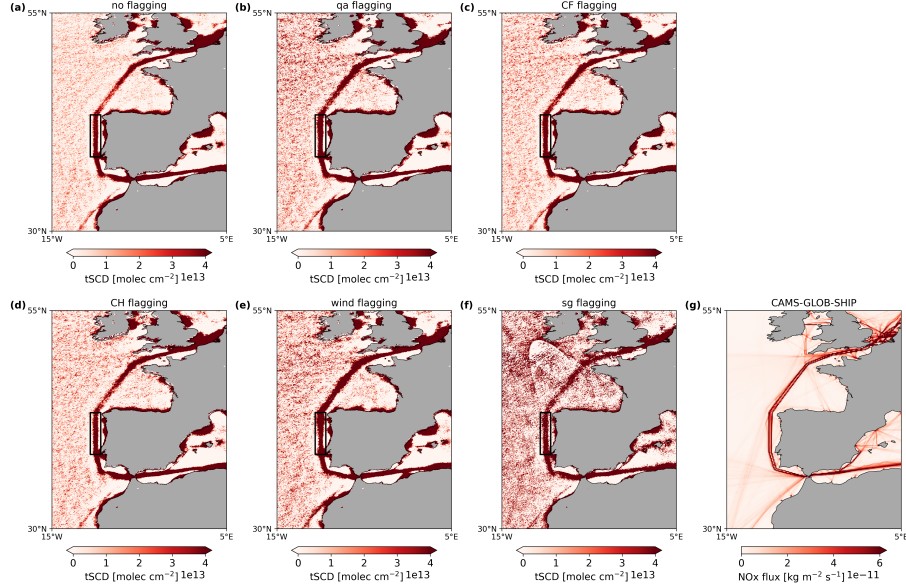

**Figure 7.** (a-f) Filtered TROPOMI and (g) CAMS-GLOB-SHIP NO$_2$ signals from shipping activity in the Atlantic Ocean near the coast of Portugal; filtered TROPOMI data with (a) no flagging, (b) qa flagging, (c) CF flagging, (d) CH flagging, (e) wind flagging, and (f) sg flagging. The rectangles highlight the area of the cross-sections shown in Fig. 8. The two separated shipping lanes visible in the filtered TROPOMI NO$_2$ data correlate well with the CAMS-GLOB-SHIP data, covering the years from 2018 to 2021.

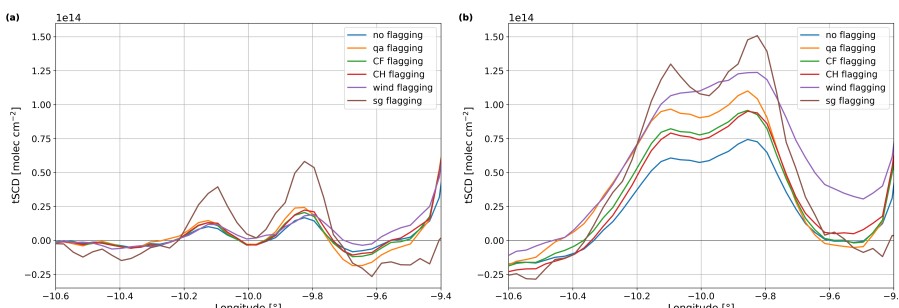

**Figure 8.** Cross-sections of the filtered TROPOMI NO$_2$ shipping lanes in the Atlantic Ocean near the coast of Portugal (rectangles in Fig. 7) for the different flagging criteria using different box sizes of approximately (a) 0.25° and (b) 1° in both longitude and latitude for the high-pass filter (corresponds to Fig. 3). The curves represent the latitudinal average from 38.5°N to 43.3°N as a function of longitude. The elevated NO$_2$ values at larger longitude (-9.4°) reflect that the NO$_2$ signal increases near land masses, here Portugal, resulting from the iterative high-pass filtering.

Examining the shipping lane in the Atlantic Ocean near the coast of Portugal, marked by the black rectangle in Fig. 7a-f, Fig. 8 displays the latitudinal average of the NO$_2$ tSCD from 38.5°N to 43.3°N, plotted as a function of longitude from 10.6°W 300 to 9.4°W for the different flagging criteria using high-pass filter box sizes of 0.25° and 1°. Using a smaller box size of 0.25°





in the high-pass filter (Fig. 8a) reveals a double-peaked pattern in the $NO_2$ signal for all flagging criteria, with the effect under the sg flagging particularly pronounced and its peaks being larger than about 2.5e13 molec cm$^{-2}$ compared to the peaks of the other curves. With a 1° box size (Fig. 8b), the $NO_2$ values of the two peaks are generally around 5e13 molec cm$^{-2}$ for the no flagging and even 9e13 molec cm$^{-2}$ for the sg flagging higher than those observed with a 0.25° box size. However,

the two peaks are less pronounced except for the ones with sg flagging, which again shows a distinct double-peak pattern and elevated $NO_2$ column levels, differing from the other peaks by approximately 2.5e13 molec cm$^{-2}$ compared to the peaks with no flagging and up to 7.5e13 molec cm$^{-2}$ compared to the wind-flagged peaks. This finding corresponds to the observations in Fig. 3, where the North-East Atlantic shipping lane is clearly separated with the smaller box size (Fig. 3a) while the $NO_2$ signal itself is much stronger with the 1° box size (Fig. 3b). The two peaks observed are attributed to ships moving in opposite

directions along the primary shipping route off the coast of Portugal, where the respective travel directions are clustered. This observation is supported by the CAMS-GLOB-SHIP emission inventory, which indicates two distinct lines within the main route (Fig. 7g).

Both the 0.25° and the 1° box sizes show elevated $NO_2$ values at -9.4° longitude (Fig. 8a and b), reaching similar values of around 0 to 5e13 molec cm$^{-2}$. These elevated $NO_2$ values result from the closeness to the Portuguese coast, where the

$NO_2$ signal increases probably due to land-based sources and is enhanced due to the high-pass filtering effect in areas near the shipping lane, like the negative $NO_2$ values, e.g., at -10.6° longitude. Notably, again, the curve with wind flagging shows, apart from the sg flagging curve, the largest $NO_2$ values and is slightly shifted. This result supports the expectation that low wind conditions would lead to $NO_2$ accumulation and amplify the signal. However, a less pronounced double-peak structure is found, especially for the 1° box size (Fig. 8b). A small gap in the southern part of the shipping signal in the cross-section

area (Fig. 7e) suggests that the reduced number of measurements due to the flagging may contribute to this effect. This signal narrowing can smear the two-peak pattern, especially when using the 1° box size, where the influence of the high-pass filter is more substantial. Comparing the results of this region to the Indian Ocean cross-sections, the curve with CH flagging shows smaller $NO_2$ values concerning the other flagging criteria. It is more similar to that flagged for CF. Given that low cloud conditions are prevalent in the North Atlantic, this is unexpected but can be explained by the higher overall cloud cover in

this region compared to the Indian Ocean. As a result, the larger number of available measurements reduces the high-pass filtering effect, smoothing the signal. The pronounced effect of sg flagging on the $NO_2$ signal can be attributed to the increased sensitivity of TROPOMI to $NO_2$ absorption near the sea surface due to the enhanced reflectivity of sunlight on the water. This also explains the more distinct double-peak pattern, as the ship plumes are better separated near the surface where sun glint has the most substantial effect. At higher altitudes, the plumes tend to mix, reducing their distinctiveness. While this

effect amplifies the $NO_2$ signal, this enhancement does not significantly improve the shipping lane detection due to increased background noise. Therefore, non-flagged data is preferred for detecting $NO_2$ emissions from global shipping lanes in this study.




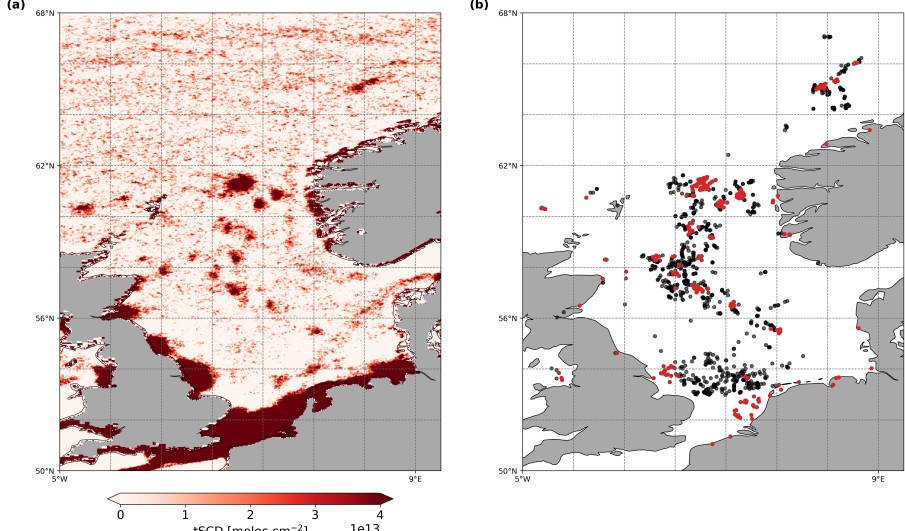

**Figure 9.** (a) Filtered TROPOMI NO$_2$ signals in the North Sea and (b) locations of oil and gas platforms (dots). The hotspot regions where the NO$_2$ signals exceed 3.5e13 molec cm$^{-2}$ (a) correlate well with locations of offshore installations, and those are indicated by the red dots (b). In other regions where oil and gas platforms are located, the NO$_2$ values show no significant enhancement, possibly due to variations in the emission rates and the high-pass filtering effect. The offshore installations are classified as "operational" in the OSPAR emission inventories for 2019 and 2021 (ODIMS, 2021).

## 3.2 Offshore platform detection

The method presented in this study enables the detection of NO$_2$ emissions not only from shipping activities but also from

offshore oil and gas platforms, which release NOx during drilling and gas flaring. Figure 9a highlights NO$_2$ hotspots in the North Sea, particularly in the northern region. These areas of elevated NO$_2$ concentrations correspond to clusters of offshore oil and gas installations documented in the OSPAR (Oslo and Paris Commission) inventory (ODIMS, 2021). The dots in Fig. 9b indicate the positions of offshore oil and gas platforms listed as operational in the OSPAR inventory for 2019 and 2021. Many of these platform locations match closely with NO$_2$ hotspots in the filtered TROPOMI NO$_2$ data; those exceeding

3.5e13 molec cm$^{-2}$ in Fig. 9a are indicated with red dots in Fig. 9b. Some larger NO$_2$ hotspot areas, such as the western section of the North Sea and north of Great Britain, overlap with only a few single offshore installations in the OSPAR data. One reason for being visible in the filtered TROPOMI NO$_2$ data might be that these platforms emit high levels of NOx due to their size or operational intensity, particularly those involved in extensive flaring or high production. Additionally, local meteorological conditions, including wind direction and temperature inversions, can trap NO$_2$ or slow its dispersion, resulting in concentrated

signals from just a few sources. Furthermore, the OSPAR inventory provides a snapshot of various platform statuses, including operational, decommissioned, closed down, unknown, derogation, dismantled, and under-construction. Consequently, there



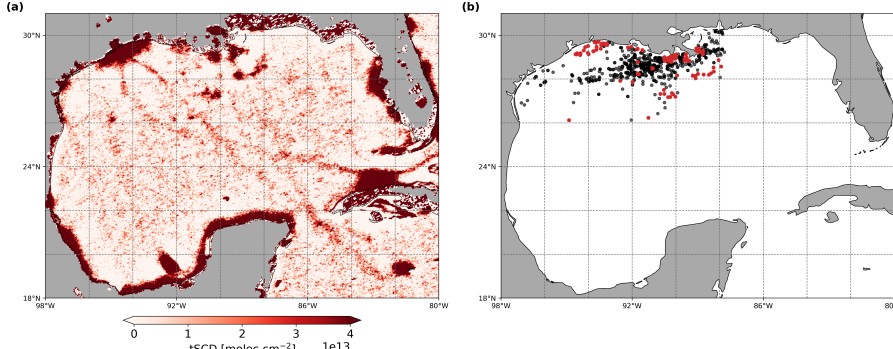

**Figure 10.** As Fig. 9, but in the Gulf of Mexico. The oil and gas platforms are listed as NOx sources in the BOEM 2021 emission inventory (BOEM, 2021).

may be gaps or delays in the OSPAR inventory regarding smaller or less permanent installations, leading to discrepancies between the detected hotspots and the documented platform locations.

Notably, the shipping lane from the Skagerrak Strait northward along the Norwegian coast is identified, with $NO_2$ signals
further intensified where this lane intersects oil and gas platform sites due to an accumulation from both the ship and platform emissions. However, in the southern North Sea, the clusters of oil and gas installations show little to no correlation with increased $NO_2$ concentrations. This discrepancy may result from significant variations in the emission rates from individual platforms; older or less active platforms may emit lower levels of $NO_2$, leading to less pronounced signals despite their presence. Additionally, atmospheric conditions should be considered, such as prevailing winds and temperature patterns, or
geographical features, including ocean currents, island formations, and proximity to land masses. These factors can significantly affect how $NO_2$ emissions are dispersed or concentrated. In some areas, wind patterns may carry $NO_2$ molecules away from their sources, reducing the concentration near the sources and causing a mismatch between high $NO_2$ concentrations and the platform locations. Furthermore, interference from other $NO_2$ sources, such as coastal cities on the eastern coast of Great Britain, possibly from wind-blown urban traffic and rural sources, may contribute to the background $NO_2$. The shipping lane
from the English Channel towards the Skagerrak Strait is overshadowed by a $NO_2$ saturation, effectively diluting the apparent impact of emissions from nearby platforms and ship lanes. These elevated $NO_2$ values result from the high-pass filtering effect near land. Finally, detection limitations may also play a significant role. Platforms with lower emissions or those spread over a larger area might fall below the detection threshold of TROPOMI, making their $NO_2$ contributions less visible.

The same phenomenon is observed in the Gulf of Mexico (Fig. 10), where numerous oil and gas platforms are situated near
the U.S. coast. Location data for these offshore installations (Fig. 10b) were obtained from the Outer Continental Shelf (OCS) Emissions Inventory, provided by the Bureau of Ocean Energy Management (BOEM) for the year 2021 (Thé et al., 2022). Not all clusters of these installations are visible in the filtered TROPOMI $NO_2$ data (Fig. 10a), potentially due to factors discussed previously, such as platform size, distance from the shore, and emission levels. At some coasts, especially where the shipping lanes from Houston enter the Gulf, the high-pass filtering amplifies the $NO_2$ values leading to circumstantial agreement of





high $NO_2$ and platform locations. However, some installations, particularly those farther offshore, unaffected by the high-pass filtering artifacts, align prominently with $NO_2$ hotspots detected in the filtered TROPOMI $NO_2$ data. This observation supports findings from Fedkin et al. (2024), who investigated $NO_2$ trends from offshore oil and gas operations in the Gulf of Mexico. Their study includes a comparable regional map that displays NOx emission hotspots from the BOEM data, which correlate well with the areas of elevated $NO_2$ values shown in Fig. 10a.

It should be noted that the elevated $NO_2$ signals detected in the southern Gulf of Mexico, particularly around the Bay of Campeche near the Mexican coast, as well as in the Caribbean Sea, are likely due to sandbanks or small, unregistered islands, rather than oil and gas platforms emissions. These signals result from the inhomogeneity effect in the TROPOMI measurements, where variability in surface reflectance amplifies the $NO_2$ signal even in the absence of emission sources. However, these results suggest that the TROPOMI data can effectively identify offshore oil and gas installations with significant $NO_2$ emissions, underscoring its value as a tool for monitoring environmental impacts from offshore operations.

## 4 Comparison between the filtered TROPOMI $NO_2$ tVCDs and the CAMS model

This study focuses on a qualitative detection of shipping $NO_2$ in TROPOMI tSCDs. However, it is of value to compare the retrieved tropospheric $NO_2$ vertical columns (tVCDs) with those from atmospheric models. For this, the $NO_2$ tVCDs from the Copernicus Atmosphere Monitoring Service (CAMS, Peuch et al. (2022)) are used. CAMS data for which a TROPOMI measurement was present are used, matching both overpass time and cloud filtering. The resulting data set has a spatial resolution of 0.4° x 0.4° and covers 5 years (2019-2023). The filtered TROPOMI $NO_2$ tSCDs are converted into tVCDs to compare the modeled CAMS data and the TROPOMI measurement. This conversion is performed by dividing the tSCDs by a constant AMF of 0.8. This method is not as precise as using pixel-specific AMF values because the AMF is affected by many factors, as discussed in Sect. 2. However, this approach provides a sufficient approximation for a simplified comparison to assess whether the shape and magnitude of the $NO_2$ distributions along selected shipping lanes differ between the modeled and observed data sets.

Figure C1a highlights prominent global shipping lanes in the CAMS data. However, the $NO_2$ saturation near land masses complicates the identification of shipping emissions in some regions. For example, in areas such as the Mediterranean Sea, shipping-related $NO_2$ emissions are not as distinct in the CAMS data compared to the filtered TROPOMI $NO_2$ data (Fig. C1b) due to the dominance of land emissions. This disparity is partially attributed to the coarser spatial resolution of the CAMS model (0.4° x 0.4°) compared to the finer grid of the filtered TROPOMI $NO_2$ data (0.03° x 0.03°). The resolution difference becomes more noticeable when examining small-scale shipping lanes such as those in the Indian Ocean. Individual pixels are conspicuous in the CAMS $NO_2$ tVCDs (Fig. 11a), whereas the filtered TROPOMI $NO_2$ tVCDs (Fig. 11b) provide more detailed spatial patterns. Despite this, both data sets detect the position of the shipping lane, as demonstrated in the cross-sections, where the peak orientations of the distributions are comparable (Fig. 11c). In the CAMS $NO_2$ tVCD retrieval, the cloud flagging considers only cloud cover. This flagging criterion is most likely comparable to the qa flagging used in the filtered TROPOMI $NO_2$ data. The CAMS data curve is offset by a value of $3.5e14 \ \mathrm{molec \ cm^{-2}}$ to align it with the magnitude



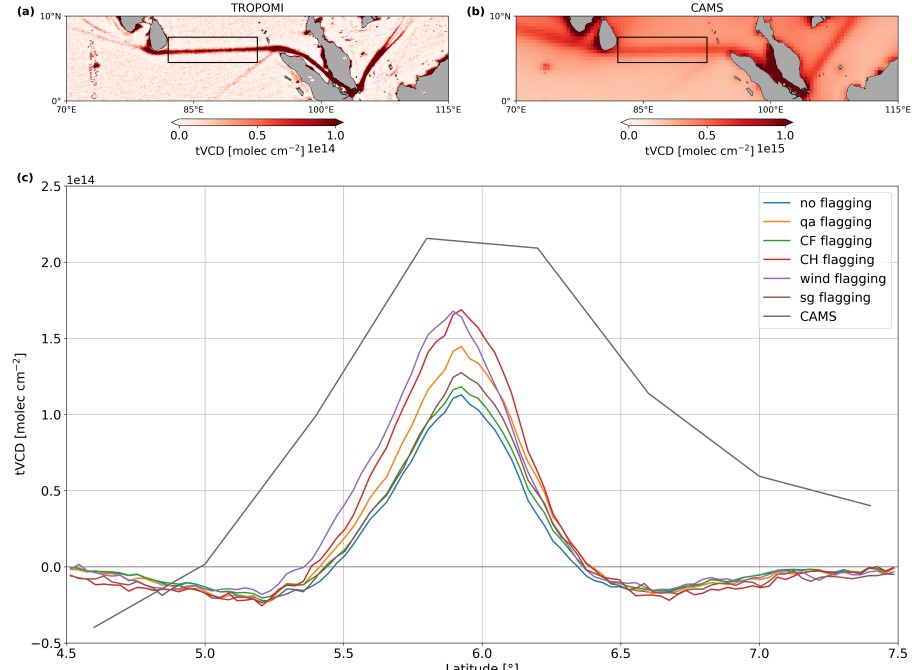

**Figure 11.** As Fig. 6, but with (a) the filtered TROPOMI NO$_2$ tVCDs, (b) the CAMS NO$_2$ tVCDs, and (c) the cross-sections of the shipping lane for the different TROPOMI flagging criteria and the CAMS data, which are offset by a value of 3.5e14 molec cm$^{-2}$. The brought-together curves of the data sets show a similar distribution peak, but the area under the CAMS tVCD curve is more extensive than under those of the filtered TROPOMI tVCDs. This broader FWHM in latitude of the CAMS curve results due to the coarser resolution of the CAMS model data, which are retrieved on a 0.4° x 0.4° grid for the years from 2019 to 2023 with cloud flagging.

of the filtered TROPOMI NO$_2$ tVCDs. However, for all that, it remains larger by approximately 7.5e13 molec cm$^{-2}$ compared to the qa-flagged TROPOMI curve. Additionally, the CAMS peak has a broader full width at half maximum (FWHM) in

latitude than the sharper peaks in the TROPOMI data.

Additional analysis of the shipping lanes near Portugal in the North Atlantic Ocean and off the coast of West Africa in the South Atlantic Ocean is provided in Appendix C (Fig. C2 and C3, respectively). While the spatial positions of the NO$_2$ signals from shipping correlate well between the modeled and measured data, the magnitude of the CAMS NO$_2$ tVCDs is significantly higher. For example, CAMS NO$_2$ values exceed the filtered TROPOMI NO$_2$ tVCDs by at least a factor of 20 for the shipping

lane peaks near Portugal (Fig. C2) and by approximately a factor of 100 for the shipping lane in the South Atlantic (Fig. C3). Possible reasons for the large differences between the TROPOMI and CAMS NO$_2$ tVCDs are that in the CAMS model, the NOx emissions from shipping are overestimated, or the chemistry in and the dilution of the ship plumes are inadequate. The latter means that, on the one hand, the photochemistry, homogeneous and heterogeneous chemistry of NOx in the CAMS model at the latitude and longitude grid of 0.4° x 0.4° does not remove enough NOx. On the other hand, the high NO$_2$ concentrations

from shipping are diluted over the size of the coarse-scale model grid cell in the CAMS model. However, in reality, the NO$_2$ is





confined to narrow plumes. This instant dilution leads to overestimated NOx concentrations from shipping over the oceans, as shown by Vinken et al. (2011).

## 5   Summary and conclusions

This study demonstrates the potential for detecting NOx emissions from global shipping routes using filtered S5P TROPOMI

$NO_2$ data. By focusing on $NO_2$ tSCD rather than tVCD, a more objective identification of NOx emissions from shipping is obtained, benefiting from the enhanced spatial resolution and avoiding the limitations of the AMFs derived from the coarse TM5 model, which inadequately captures localized emissions. The preprocessing methods, such as smoothing of the stratospheric field and iterative high-pass and Fourier filtering, significantly improve the detection of global shipping lanes in the TROPOMI $NO_2$ data and enable shipping lanes previously undetectable in satellite observations or obscured in AIS-tracked data to be

identified.

The influence of the box size in the high-pass filter on interpreting TROPOMI $NO_2$ data is examined. Smaller box sizes enhance the visibility of narrow features like the separated shipping lanes in the North Atlantic or several more minor shipping routes in the Mediterranean Sea. In contrast, larger box sizes produce overall stronger $NO_2$ signals. This study identifies a 1° box size as appropriate for balancing $NO_2$ signal detection with minimized background distortion, but other choices may be

more appropriate for local studies. The high-pass filtering also emphasizes inhomogeneity effects on $NO_2$ signals, especially in regions with varied surface reflectance near coastlines, sea ice, and islands. These surface inhomogeneities result in artificially elevated $NO_2$ values, as contrasting surface types affect reflectivity. Coastal and polar regions, in particular, exhibit these artifacts, complicating accurate emission source identification.

Several prominent TROPOMI $NO_2$ signals attributable to major shipping lanes, such as in the Red and Arabian Seas, the

Gulf of Mexico, and the Seas of Asia, are successfully detected in the filtered TROPOMI data. Many additional identified shipping routes have never been shown in satellite $NO_2$ data before. The high-resolution images can be browsed at https://www.iup.uni-bremen.de/doas/tropomi_ships.htm. The $NO_2$ signals reveal a strong correlation with shipping activities, as confirmed by the CAMS-GLOB-SHIP inventory. For the first time, the two separated shipping lanes near the coast of Portugal and towards the British Channel could be identified in satellite $NO_2$ data. This study also detected unknown shipping routes,

for example, in the Bering Sea, where AIS data may be incomplete due to disabled trackers, highlighting the limitations of traditional AIS monitoring in capturing all shipping activities, particularly for military and unauthorized vessels operating in sensitive regions.

Moreover, this study analyzes the impact of flagging criteria, such as quality value, cloud fraction, cloud height, wind speed, and sun glint, on the $NO_2$ signal distribution along shipping lanes in the Indian Ocean and off the coast of Portugal.

The cross-sections of the shipping route from Sri Lanka to the Malacca Strait reveal a pronounced $NO_2$ peak, with signal intensity varying depending on the applied flagging criteria. Non-flagged data produce the lowest $NO_2$ peak, while flagging generally increases the peak intensity at the lane's center. The strongest signals occur under low cloud heights, attributed to increased reflectivity and TROPOMI's enhanced sensitivity to $NO_2$ near the surface. Low wind speeds also influence the signal,



causing an increase and slight shift in the peak location, likely due to NO$_2$ accumulation in calm conditions. These findings
underscore the importance of selecting appropriate flagging criteria when using satellite data to monitor NOx emissions, as
each criterion can significantly influence the signal strength and the number of available measurements. In the shipping route
off the coast of Portugal, a consistent double-peak NO$_2$ pattern is observed across all flagging criteria, particularly when
using a smaller box size of 0.25° in the high-pass filter. This pattern corresponds to the structure of ships traveling in two
directions along the main route, as confirmed by the CAMS-GLOB-SHIP inventory. Although a larger box size of 1° reduces
the visibility of the double-peak pattern, it amplifies the overall NO$_2$ signal for all flagging criteria. The application of sun
glint flagging further enhances the NO$_2$ signal due to increased surface reflectivity. However, this also diminishes overall lane
detection effectiveness by increasing background noise from reduced measurement availability. Consequently, non-flagged
data are preferred for presenting global NO$_2$ emissions from shipping, as they provide the best balance between signal clarity
and noise minimization.

Furthermore, this methodology enables the detection of NOx emissions from offshore oil and gas platforms, as shown by
comparing TROPOMI data with emissions inventories, such as OSPAR in the North Sea and BOEM in the Gulf of Mexico.
This positive correlation demonstrates that TROPOMI can effectively capture emissions from offshore operations across different regions. Additionally, TROPOMI data highlight areas in the North Sea where shipping lanes and oil platforms intersect,
resulting in intensified NO$_2$ concentrations due to the cumulative effect of emissions from both sources. Factors such as the
high-pass filtering can influence the visualization of the TROPOMI NO$_2$ signals, sometimes obscuring emissions or amplifying
contrasts in areas with high background NO$_2$ concentrations. While acknowledging these limitations, this study confirms the
potential of TROPOMI data as a valuable tool for tracking NO$_2$ emissions from offshore sources.

Finally, an initial comparison between the CAMS model and the filtered TROPOMI NO$_2$ tVCDs shows that both data sets
effectively identify the positions of global shipping lanes. However, the CAMS data exhibits significantly higher NO$_2$ tVCD
values, exceeding the filtered TROPOMI measurements by factors of approximately 6 in the Indian Ocean, 20 near Portugal,
and 100 in the South Atlantic Ocean. The coarser resolution of CAMS (0.4°x0.4°) limits its ability to resolve small-scale spatial
features of shipping lanes, which are better captured by the finer resolution of the filtered TROPOMI data (0.03° x 0.03°).
Furthermore, the high NO$_2$ saturation near land masses in the CAMS data, probably due to the mixing of land-based and
shipping emissions, reduces the precision in regions such as the Mediterranean Sea. The reason for the large difference between
the retrieved NO$_2$ tVCDs from TROPOMI and those simulated by the CAMS model, which is a factor in the range 20 to 100
larger, is not yet explained. Possible reasons for the differences are i) overestimates of the NOx emissions from ships or ii) an
inadequate description of the loss of NOx in the plume of the ship emission in CAMS. The latter probably results from the
fact that CAMS simulations dilute emissions over the entire grid cell, whereas shipping plumes are much narrower, requiring a
plume chemistry approach rather than a chemistry transport model approach. Further research is needed to explain this effect
on global shipping-related NO$_2$ emissions in detail.

In conclusion, this study presents methodological advancements for analyzing NOx emissions from shipping using TROPOMI
data. Through refining data preprocessing, understanding the effects of the high-pass filtering and inhomogeneous surfaces, and
carefully selecting flagging criteria, we have improved the detection capability for NO$_2$ emissions from shipping and offshore



activities. This approach also establishes a foundation for future research on the global environmental impacts of maritime
traffic, supporting international regulatory efforts to monitor and reduce NOx emissions from shipping.

*Data availability.*  TROPOMI $NO_2$ data from May 2018 onward are publicly available through the Copernicus Data Space Ecosystem (https: //dataspace.copernicus.eu). CAMPS-GLOB-SHIP data are accessed via the ECCAD Catalogue (https://permalink.aeris-data.fr/CAMS-GLOB-SHIP). Information on oil and gas platform locations is available for OSPAR data through ODIMS at https://odims.ospar.org/en/search/?datastream= offshore_installations and for BOEM data at https://www.boem.gov/environment/environmental-studies/2021-ocs-emissions-inventory. CAMS
global model data are based on modified Copernicus Atmosphere Monitoring Service Information 2024; neither the European Commission nor ECMWF is responsible for any use that may be made of the information this publication contains. High-resolved global maps of the filtered TROPOMI $NO_2$ tSCDs for different box sizes in the high-pass filter can be explored at the University of Bremen IUP DOAS website: https://www.iup.uni-bremen.de/doas/tropomi_ships.htm.





**Appendix A: Details on the filtering methods**

The data analysis and visualization in this study were performed using Python, including the version 1.10.1 of the "SciPy" library for the high-pass and Fourier filtering techniques. "SciPy" offers a scientific collection of mathematical algorithms.

**A1   High-pass filtering**

The Python function "generic_filter" from the multidimensional image processing package "scipy.ndimage" is used to compute the rolling mean, utilizing an averaging function with the mode parameter set to 'constant'. This configuration extends the input array by filling values beyond the boundaries with the constant 'NaN' (Not a Number). This study defines the standard box size as 33 pixels in both longitude and latitude, equivalent to approximately 1°, based on a pixel size of 0.03° in the gridded $NO_2$ tSCD data. As noted earlier, the box size can be adjusted for specific regions, e.g., with the 0.25° box size including 9 pixels in the rolling mean calculation.

After calculating the rolling mean values, $rolling\_mean$, they are subtracted from the original gridded $NO_2$ tSCD pixels, $no2\_tscd$, located at the center of each rolling mean box. This calculation yields the high-pass filtered $NO_2$ tSCD pixels as: $filtered\_no2\_tscd = no2\_tscd - rolling\_mean$.

**A2   Fourier filtering**

The Python package "scipy.fftpack" for fast Fourier transform (FFT) is used for the Fourier filtering method. Each latitudinal pixel row is considered individually for the FFT analysis in this processing to mitigate the stripe-like pattern oriented in the direction of the TROPOMI orbits. In the first step, all pixels containing $NO_2$ tSCDs that are larger than 1e14 molec cm$^{-2}$ or smaller than -1e14 molec cm$^{-2}$ are set to 'NaN'. These include, in particular, the highest and lowest values on coastlines due to the inhomogeneity and high-pass filter effects. Otherwise, these pixels would influence the FFT signal too strongly. Next, the latitudinal row is truncated at all 'NaN' values into shorter segments to consider only valid values. This step is necessary because 'NaN' values cannot be considered in an FFT analysis. After these preparations, the Python function "scipy.fftpack.fft" is applied to the individual data segments to calculate the discrete Fourier transform (DFT) using the efficient FFT algorithm (Cooley and Tukey, 1965):

$$y[k] = \sum_{n=0}^{N-1} \exp\left(-2\pi j \frac{kn}{N}\right) x[n] \tag{A1}$$

where $y[k]$ is the FFT of length $N$ of the sequence $x[n]$. The FFT sampling frequency points are calculated using the Python function "scipy.fftpack.fftfreq". Several tests showed that the following approach is the most suitable for mitigating the orbit structures in the high-pass filtered TROPOMI $NO_2$ tSCDs: The FFT frequencies between 20.67 and 24 cycles per ° and between 41.67 and 45 cycles per ° are averaged, and the mean values are used as the values in these frequency ranges. As a result, the periodical stripe-like pattern significantly disappears. After the frequency filtering of the FFT, the inverse of the DFT





is computed using the Python function "scipy.fftpack.ifft", which is defined as follows:

$$x[n] = \frac{1}{N} \sum_{k=0}^{N-1} \exp\left(2\pi j \frac{kn}{N}\right) y[k] \tag{A2}$$

525   In the last step, each latitudinal row is again concatenated, corresponding to the longitudes, with all 'NaN' values, the excluded highest/lowest values, and these Fourier-filtered signal components to reconstruct the original data structure, ensuring that the filtered TROPOMI NO$_2$ data can be used for the analysis and visualization.



## Appendix B: Additional maps of ship emissions in the filtered NO$_2$ TROPOMI data

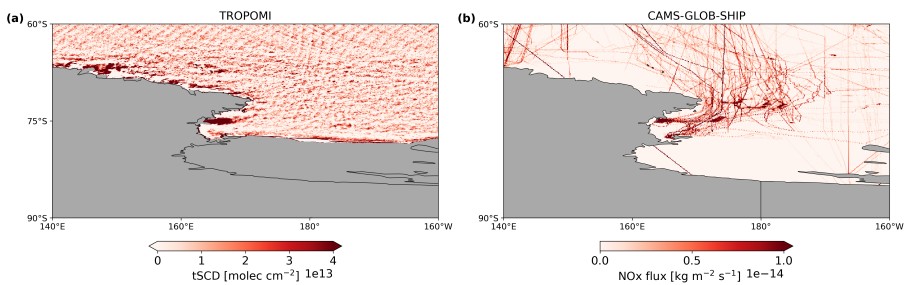

**Figure B1.** The higher NO$_2$ signal in the (a) filtered TROPOMI data in the Southern Ocean near the Antarctic continent can be assigned to (b) CAMS-GLOB-SHIP NO$_2$ shipping activities in the Ross Sea. The CAMS-GLOB-SHIP data cover the years from 2018 to 2021. The enhanced NO$_2$ signal is detected only in the summer season with minimal sea ice extent, while during winter, the NO$_2$ signal in the Antarctic region is dominated by sea ice cover, indicating that these NO$_2$ emissions may originate from ship emissions.

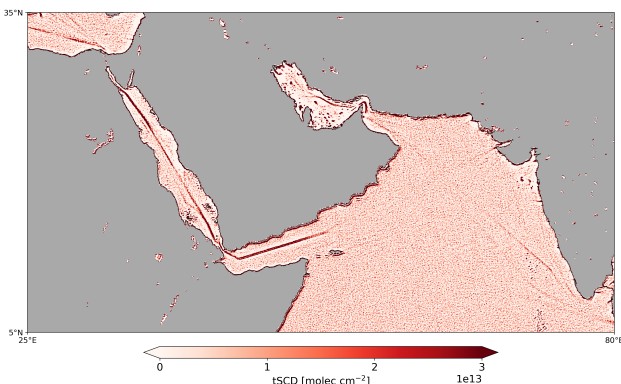

**Figure B2.** Filtered TROPOMI NO$_2$ signals from shipping activity in the Red and Arabian Seas with the non-iterative high-pass filtering at a box size of approximately 0.25° in both longitude and latitude. Two separated shipping lanes are visible in the Gulf of Aden. In the Persian Gulf, a distinct shipping route is visible near the coast of Iran, and the dot-shaped elevated NO$_2$ values indicate NOx emissions from offshore platforms. However, the shipping NO$_2$ signals in the Arabian Sea are not as pronounced as in the filtered TROPOMI data using the iterative approach with a box size of 1° (see Fig. 4a).



**Appendix C: Additional comparisons between the filtered TROPOMI NO$_2$ data and the CAMS model**

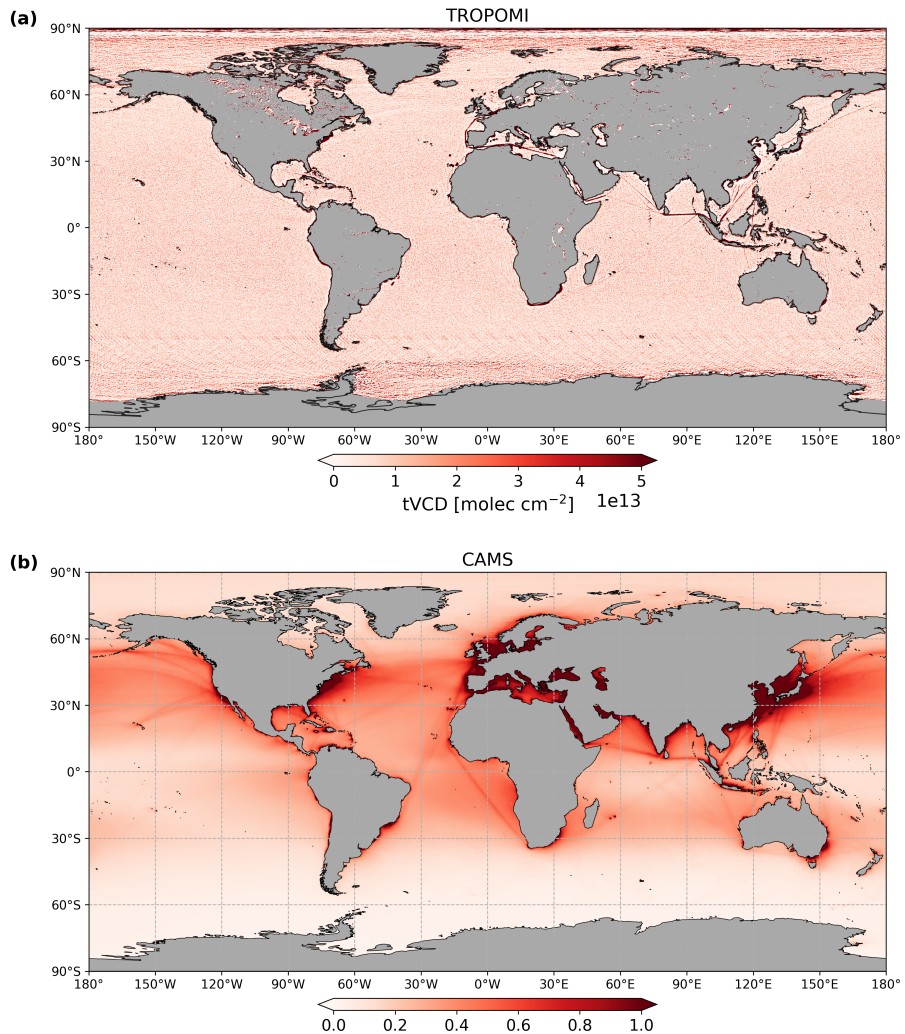

**Figure C1.** On a global scale, the magnitude of (a) the filtered TROPOMI NO$_2$ tVCDs is, on average, at least a factor of 10 smaller than the magnitude of (b) the CAMS NO$_2$ tVCDs. Additionally, differences are found in the spatial extent of the shipping lanes, resulting from the coarser resolution of the CAMS model data, which are retrieved on a 0.4° x 0.4° grid from 2019 to 2023 with cloud flagging. However, the positions of a great number of shipping routes, visible in the CAMS data, are identified in the filtered TROPOMI NO$_2$ data. In addition, the CAMS model data show a higher NO$_2$ signal in the Ross Sea near the Antarctic continent at a similar position to the filtered TROPOMI data (see also Fig. B1a), which confirms the assumption that this is an actual NO$_2$ emission signal.





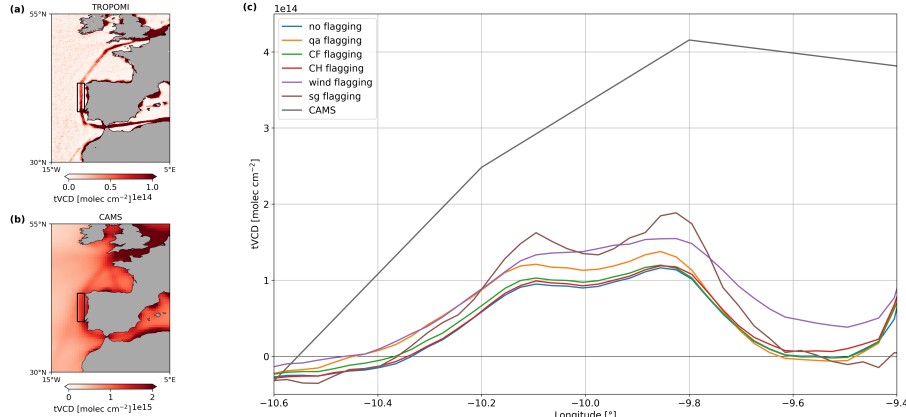

**Figure C2.** (a) Filtered TROPOMI NO$_2$ tVCDs, (b) CAMS NO$_2$ tVCDs, and (c) the cross sections of the shipping lane in the North Atlantic Ocean near the coast of Portugal (black rectangles) for the different TROPOMI flagging criteria and the CAMS data, which are offset by a value of 1e15 molec cm$^{-2}$. Besides the at least 20 times larger CAMS NO$_2$ values compared to the TROPOMI data, the two peak patterns of this small-scale shipping lane are not as clearly identified in the CAMS curve as in the TROPOMI tVCDs due to the coarser resolution of the CAMS model data. In addition, this poor resolution leads to a mixing of shipping signals and emissions from land, here visible in the considerable NO$_2$ value at -9.4° longitude, which is not clearly set aside from the signal peak at -9.8° longitude.

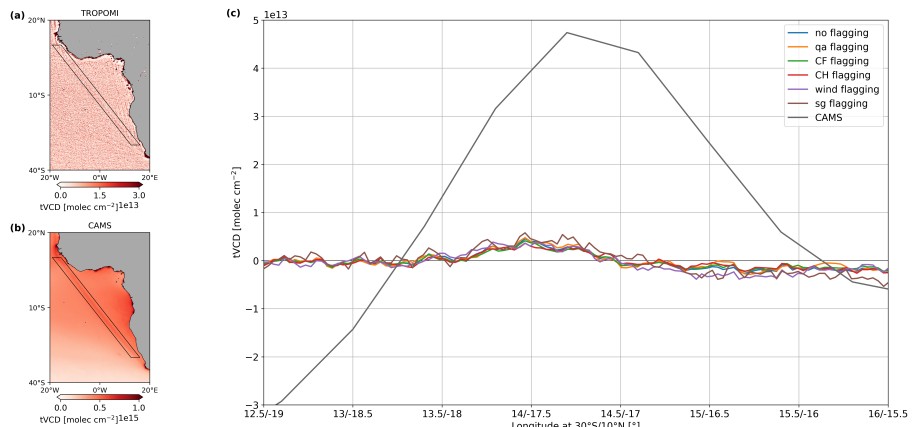

**Figure C3.** As Fig. C2, but (c) presents the cross-sections of the black polygons of the shipping lane in the South Atlantic Ocean west of the African continent. The CAMS NO$_2$ tVCDs are offset by a value of 4.5e14 molec cm$^{-2}$, and thus, capture the shipping lane with approximately 100 times larger values than the filtered TROPOMI NO$_2$ tVCDs, which do not exceed 0.5e13 molec cm$^{-2}$ at the maximum peak. The filtered TROPOMI NO$_2$ curves vary greatly in this region due to the S5p orbit-oriented structures enhanced by the high-pass filtering, while the CAMS peak is straight-lined due to the coarser resolution. The higher values on the right side of the CAMS curve result from larger signals near the continent, possibly due to emission outflow. In comparison, the smaller values on the left curve side correspond to weaker signals in the open ocean.





530 *Author contributions.* ML and AR designed the study. ML performed the data analysis and wrote the manuscript with contributions from AR, JPB, and HB.

*Competing interests.* At least one of the (co-)authors is a member of the editorial board of Atmospheric Measurement Techniques.

*Acknowledgements.* We thank Dr. Anja Schönhardt for her help with the CAMS model data. Parts of this work were funded by the Deutsches Zentrum für Luft- und Raumfahrt (DLR) under contract 50EE1811A ("S5P Datennutzung"), and the University of Bremen.



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
