# Peer review of "Improved detection of global $NO_2$ signals from shipping in Sentinel-5P TROPOMI data"

_EGUsphere, 2025_

## Referee Comment (RC1)

**Review:**

**Latsch, et al. Improved detection of global NOx emissions from shipping in Sentinel-5P TROPOMI data.**

**Summary:**

The authors present a sophisticated approach for detecting persistent pollution tracks along shipping lanes, using careful filtering of combined 6 years of TROPOMI $NO_2$ data. The resultant maps clearly reveal multiple tracks in each of the earth's oceans. The study is noteworthy in its detailed presentation of the methods used and the detection of tracks not previously identified in satellite data or that may otherwise be unknown. The authors test a variety of approaches, describing the pros and cons of each and perform quantitative model comparisons. I have only a few minor suggestions for clarification and possible modification of the filtering techniques that might overcome some of the shortcomings pointed out by the authors.

Overall, the authors have done due diligence in rigorous testing of their algorithm and particularly in their thorough analysis of the sources of the tracks visible in their final product. I believe a version of the paper close to its present form is worthy of publication. Below are a few minor corrections and suggestions.

**Comments:**

(1) There is a tradeoff between capture of fine detail, enhancement of weaker signals and suppression of coastline artifacts. Might a dynamic box size – smaller near coastlines and larger over open waters – mitigate some of these concerns? For a given track, the enhancement would also vary with box size, but this approach would allow all tracks to be captured in a single map. A track out at sea could be followed to and along a coastline without being lost in the clutter. SciPy generic_filter does not directly support spatially varying footprint sizes, but it should be straightforward to develop code for this. Depending on the difficulty of implementing this in the present study, it would be interesting to see an approach like this tested over a small region.

(2) Line 36: "Satellite retrievals of tropospheric…"

(3) Line 120: "…are used to high-pass filter…"

(4) Lines 138-139: CH flagging selects both clear scenes and cloudy scenes with low clouds. Is there a cloud-fraction threshold for the cloudy scenes?

(5) Line 182: "In the northern hemisphere, the higher $NO_2$ signal is detected only when the sea-ice extent is minimal (June – November), while during the winter season…"

(6) Lines 257-258: "…to investigate the individual impacts of various flagging criteria…"

(7) Line 323: "…smaller $NO_2$ values than the other criteria…"

(8) Lines 322-326: The differences in peak heights for CH flagging between the Indian Ocean and the Atlantic are attributed to the greater number of cloudy cases in the Atlantic. Is this because CH flagging, in general, includes both clear and low-altitude cloudy scenes?

(9) Line 388: Please briefly mention why the AMF of 0.8 was chosen. It is a reasonable value, but obviously the AMF can depend strongly on whether clouds are present, etc.

(10) Line 392: "Figure C1b…CAMS data"

(11) Line 394: "…TROPOMI $NO_2$ data (Fig. C1a)…"

(12) Line 398: "…TROPOMI $NO_2$ tVCDs (Fig. 11a)…"

(13) Line 416:  Should the sentence read "Inadequate dilution in CAMS could lead to overestimated NOx…" ?

---

## Author Comment (AC1)

**EGUSPHERE-2025-107 Author's comment to RC1 by Anonymous Referee #4 (Miriam Latsch et al.)**

Legend: Referee comments in **black**, author comments in **blue**, changes in the manuscript text in **green**

Summary:

The authors present a sophisticated approach for detecting persistent pollution tracks along shipping lanes, using careful filtering of combined 6 years of TROPOMI NO2 data. The resultant maps clearly reveal multiple tracks in each of the earth's oceans. The study is noteworthy in its detailed presentation of the methods used and the detection of tracks not previously identified in satellite data or that may otherwise be unknown. The authors test a variety of approaches, describing the pros and cons of each and perform quantitative model comparisons. I have only a few minor suggestions for clarification and possible modification of the filtering techniques that might overcome some of the shortcomings pointed out by the authors.

Overall, the authors have done due diligence in rigorous testing of their algorithm and particularly in their thorough analysis of the sources of the tracks visible in their final product. I believe a version of the paper close to its present form is worthy of publication. Below are a few minor corrections and suggestions.

We want to thank Referee #4 for the positive feedback, helpful comments, and advice on our manuscript. Detailed responses to the reviewer's comments can be found below. We hope that we have incorporated all suggestions and comments in a satisfactory way. At the end of this document, you will find a general section on additional changes we made during the review process.

Comments:

(1) There is a tradeoff between capture of fine detail, enhancement of weaker signals and suppression of coastline artifacts. Might a dynamic box size – smaller near coastlines and larger over open waters – mitigate some of these concerns? For a given track, the enhancement would also vary with box size, but this approach would allow all tracks to be captured in a single map. A track out at sea could be followed to and along a coastline without being lost in the clutter. SciPy generic_filter does not directly support spatially varying footprint sizes, but it should be straightforward to develop code for this. Depending on the difficulty of implementing this in the present study, it would be interesting to see an approach like this tested over a small region.

Thank you for this idea which we included in the manuscript as an additional test. We added a discussion of the advantages and disadvantages of this approach as follows to our manuscript in Section 3, starting in Line 201:
"Nevertheless, another approach was tested, using two different box sizes depending on the distance from the coastline. This method of individual box sizes allows the

implementation of the smaller box size of 0.25° for pixels close to the coastline, while the standard box size of 1° is used for all other pixels that are further away from the coastal pixels and in the open oceans. The advantages and disadvantages of this method are discussed below for a selected region, the seas around Europe, Africa and Asia (see Fig. B2). Figure B2b shows the $NO_2$ signals of the different box sizes with a coastal area of 50 pixels (~1.5°) from the shoreline, within which the smaller box size is used. In this case, the shipping lanes in narrow ocean regions such as the Persian Gulf are well resolved, the separated shipping routes in the Atlantic Ocean and the English Channel are visible, and the large $NO_2$ values near the coast caused by the scene inhomogeneity and high-pass filter effects are removed. However, the shipping lanes near land pixels that have the largest $NO_2$ values at a box size of 1°, such as the busy shipping route in the Indian Ocean from Sri Lanka to Indonesia and the shipping lanes in the Mediterranean Sea, are reduced or even canceled due to the smaller box size resolution, as in the Black Sea. Therefore, the effect of a smaller coastal region of only 17 pixels (~0.5°) on the $NO_2$ signals is also investigated (Fig. B2c). Under this specification, the well-defined shipping lanes near the coast are displayed clearly and without interruptions. However, this coastal region is not wide enough to visualize the small-scale $NO_2$ signals in narrow regions such as the Persian Gulf. The main advantage of this approach is that the artifacts of high $NO_2$ values near the coast caused by the high-pass filtering are compensated for in most regions, regardless of the width of the coastal region. However, the effect of this method on the visualization of shipping lanes largely depends on the region and its proximity to the coast. In some areas, such as the southeast of South Korea and Japan, inserting the values of the smaller box size close to the coast separates the artefacts from the land and thus could be perceived as shipping signals. The bottom line is that this method can significantly improve the detection of shipping signals and reduce background levels for specific regions, but its results are not consistent on a global scale. Therefore, the following analysis uses the standard box size of 1° to provide an overview of our method, which can be applied globally."

The corresponding Figure, which displays the $NO_2$ signals using the two box sizes for two different coastal region thresholds of 50 and 17 pixels, is added to the Appendix.

(2) Line 36: "Satellite retrievals of tropospheric…"

The beginning of this sentence has been changed as suggested.

(3) Line 120: "…are used to high-pass filter…"

Thank you. "filtering" has been changed to "filter".

(4) Lines 138-139: CH flagging selects both clear scenes and cloudy scenes with low clouds. Is there a cloud-fraction threshold for the cloudy scenes?

The CH flagging includes cloudy scenes with a cloud height of less than 2 km and situations in which no clouds are present, as no threshold value for the cloud fraction is

defined for this flagging criterion. We thought this was explained in the sentence in lines 138-140; however, we have changed it slightly to clarify this setting:

"The cloud height (CH) flagging uses only pixels without clouds (clear-sky scenes) and with clouds at altitudes below 2 km (with no CF threshold) to avoid higher clouds that may shield the shipping $NO_2$ from the satellite view."

See also our reply to comment (8), where we discuss other settings for the CH flagging.

(5) Line 182: "In the northern hemisphere, the higher NO2 signal is detected only when the sea-ice extent is minimal (June – November), while during the winter season…"

Thank you for your suggestion. Here, we have examined the Antarctic region but incorrectly assigned the summer and winter months of the southern hemisphere. The sentence has been changed to:

"In the Ross Sea, the higher $NO_2$ signal is detected only when the sea ice extent is minimal (from December to April), while during the winter season (from June to November), the characteristic sea ice pattern dominates the TROPOMI $NO_2$ data in the polar region."

(6) Lines 257-258: "…to investigate the individual impacts of various flagging criteria…"

Thank you, this has been implemented.

(7) Line 323: "…smaller NO2 values than the other criteria…"

Thank you, this has been implemented.

(8) Lines 322-326: The differences in peak heights for CH flagging between the Indian Ocean and the Atlantic are attributed to the greater number of cloudy cases in the Atlantic. Is this because CH flagging, in general, includes both clear and low-altitude cloudy scenes?

Thank you for your comment. The CH flagging includes both cloudy scenes with low clouds and clear-sky scenes, as no cloud fraction threshold is defined. When distinguishing between cloudy and clear-sky scenarios, the CH-flagged curves in the Indian and Atlantic Oceans behave differently, as discussed in the following.

The CH-flagged cross-sections of the shipping lanes near the coast of Portugal change for different settings, just as we expected (Figure 1). When the predominantly cloud-free scenes (CF<0.1) are considered in addition to those measurements having cloud heights below 2 km (Figure 1b), the CH-flagging curve is higher than the curve that has only the CH threshold (Figure 1a). This fact can be explained by clouds shielding $NO_2$ from the satellite, as shipping $NO_2$ is located below the clouds. As a logical consequence, the $NO_2$ column is larger in clear-sky scenes, as $NO_2$ can be detected at all altitudes. If the clear-sky scenes are excluded with a CF threshold of 0.1 (Figure 1c) or a stricter CF threshold of 0.2 (Figure 1d), the $NO_2$ columns are lower than that in Figure 1a. Here, the scenes where clouds shield

NO₂ dominate, resulting in a generally smaller NO₂ column. These results may indicate that NO₂ is trapped at low altitudes in this region.

[Figure]

*Figure 1: Testing different settings of the CH flagging for the shipping lanes in the Atlantic Ocean; the other flagging criteria are unchanged. (a) The CH flagging is only defined by a cloud height threshold of 2 km. (b) CH threshold as before, but mainly clear-sky scenes are included with a CF threshold of 0.1. (c) CH threshold as before, but clear-sky scenes are excluded with a threshold of 0.2. (d) CH threshold as before, but the CF threshold included more slightly cloudy scenes with a threshold of 0.1, thus, clear-sky scenes are still excluded.*

In contrast, the cross-sections of the Indian Ocean shipping lane show an entirely different behavior when comparing the various settings for the CH flagging. When only clear-sky scenes are included (Figure 2**Fehler! Verweisquelle konnte nicht gefunden werden.**b), the peak of the CH-flagged curve is comparable in height to that of the curve without a CF threshold (Figure 2**Fehler! Verweisquelle konnte nicht gefunden werden.**a). A significant difference between the two regions is that the CH-flagged curve, together with the wind-flagged curve, has the highest NO₂ values compared to the other flagging criteria. If the clear-sky scenes are excluded, i.e. only situations with cloud cover are considered (Figure 2c and d), the NO₂ columns of the CH-flagged curves are even higher than with other flagging settings and exceed the previously defined y-axis scale. Figure 2c and, especially, Figure 2d show many oscillations in the CH-flagged curve due to the significantly reduced number of measurements within this CH flagging, indicating that clear-sky scenes dominate in this region. These high NO₂ peaks in cloudy scenes might be because some of the NO₂ is located above the clouds, which, as the reflectivity is increased, leads to higher NO₂ slant columns. This could be explained by the fact that the clouds are at lower altitudes in the Indian Ocean than in the Atlantic Ocean or that there is more vertical mixing over the warm Indian Ocean.

[Figure]

*Figure 2: As Figure 1, but for the shipping lane in the Indian Ocean.*

Because we only want to give an overview of the impacts of the various flagging criteria on the $NO_2$ column and do not want to go into these details, we decided to change only the sentence in lines 138-140 slightly to clarify the definition of the CH flagging, as stated in the answer to your comment (4).

Furthermore, we updated the sentences in Lines 280-284 regarding the Indian Ocean: "Flagging for the lowest CHs is among the most effective criteria due to the increased $NO_2$ peak. In addition to the fact that this flagging criterion includes the clear-sky scenes, one potential reason for this higher peak is that low clouds can reflect more sunlight towards the satellite, enhancing the sensitivity of the TROPOMI instrument for $NO_2$ above the clouds. This increased sensitivity may lead to higher slant columns of $NO_2$ in the shipping lanes when only the lowest CHs and clear-sky scenes are considered."

In addition, we changed the sentences in Lines 322-326 to clarify why the CH-flagged curves behave differently in the two regions:

"Comparing the results of this region to the Indian Ocean cross-sections, the curve with CH flagging shows smaller $NO_2$ values than with the other criteria. It is more similar to that flagged for CF. As low cloud conditions are prevalent in the North Atlantic, the clear-sky scenes do not dominate the curve with CH flagging. Hence, this lower peak can be explained by the fact that the $NO_2$ in this region is predominantly below the clouds, unlike in the Indian Ocean, where some of the $NO_2$ is presumably present above the clouds. As a result, the clouds shield the $NO_2$ in the Atlantic Ocean, resulting in smaller $NO_2$ slant columns."

(9) Line 388: Please briefly mention why the AMF of 0.8 was chosen. It is a reasonable value, but obviously the AMF can depend strongly on whether clouds are present, etc.

The AMF of 0.8 was chosen as a simple value to be applied to the averaged and filtered slant columns. In a quantitative analysis, the AMF would have to be scene specific and would have to account for varying surface reflectance, clouds, aerosols and different vertical profiles of $NO_2$. Here, we assumed a surface albedo of 0.04, no aerosols and a vertical mixing of a few hundred meters (see Figure below).

[Figure]

(10) Line 392: "Figure C1b…CAMS data"

(11) Line 394: "…TROPOMI NO2 data (Fig. C1a)…"

(12) Line 398: "…TROPOMI NO2 tVCDs (Fig. 11a)…"

Thank you for pointing that out. The letters of the referenced figure numbers have been corrected for comments (10), (11) and (12), as well as that in Line 398, which is now "CAMS NO2 tVCDs (Fig. 11b)".

(13) Line 416: Should the sentence read "Inadequate dilution in CAMS could lead to overestimated NOx…" ?

Thank you for this suggestion. The (inadequate) dilution approach in CAMS is commonly described by the term "instant dilution", which is also used in the study by Vinken et al. (2011). Therefore, we adopted this term in our manuscript. The sentence has been changed to:

"This instant dilution approach in CAMS could lead to overestimated NOx concentrations from shipping over the oceans, as shown by Vinken et al. (2011)."

**Other changes made to the manuscript during the review process:**

All figures with maps (Figures 2-7, 9-11, B1-B2, and C1-C3) have been changed: The inland water pixels are now also flagged as NaN values to focus on the ocean regions and to reduce the scatter over the continents.
Additionally, Figure B1 and Figure C1: The surface classification mask of the TROPOMI data is now also applied to the CAMS data. Thus, the same pixels are flagged as NaNs for visual consistency. The reason why these water pixels are marked as invalid is that the TROPOMI surface classification mask defines snow and ice pixels over water as land pixels. In this study, only water pixels are included. However, these flagged pixels are irrelevant for this study because they represent the snow and ice edge and, therefore, are not affected by ship emissions.

We added the following sentence to the Acknowledgments:
"We thank the two anonymous reviewers for their comments and suggestions which helped to improve the results and their presentation in this manuscript."

---

## Author Comment (AC2)

**EGUSPHERE-2025-107 Author's comment to RC2 by Anonymous Referee #1 (Miriam Latsch et al.)**

Legend: Referee comments in **black**, author comments in **blue**, changes in the manuscript text in **green**

The manuscript presents a worldmap of patterns of enhanced NO2 over oceans that can be attributed to ship tracks or oil rigs, based on high-pass filtered NO2 measurements from TROPOMI. The paper is well generally well written, and the resulting ship tracks are impressive by the presented detail.

We want to thank Referee #1 for the feedback and helpful comments on our manuscript. Detailed responses to the reviewer's comments can be found below. We hope that we have incorporated all suggestions and comments in a satisfactory way. At the end of this document, you will find a general section on additional changes we made during the review process.

I have the following general concerns:

- The paper presents patterns in a rather qualitative way (tSCDs), while in the title as well as in some instances in the text it sounds as if the study aims to quantify ship emissions. This should be avoided.

We agree with this comment and in response the title has been changed (see below). In addition, we have changed the term "NOx emissions" to "NO$_2$ signals" in Lines 419, 460, and 481. Furthermore, we have added the term "qualitative" or "qualitatively" to Lines 5, 55, 419, 460, and 481.

- The methodology should be clarified and the individual retrieval/filtering steps should be illustrated for real measurements.

Done; see below for details.

- The comparison to CAMS VCDs is problematic, as this dataset has not been high-pass filtered. This should be modified, or discussed properly.

Thank you for this hint. We discussed this in detail below.

- The outlook/next steps should be discussed in more depth. Obviously, it would be desirable to quantify NOx emissions for the detected ship tracks. How to do this, which problems have to be solved? (impact on filter settings, AMFs, ...)

Thank you for this comment. We have added the following sentence to Line 485:
"The next step for quantitative estimation of NOx emissions for the detected shipping tracks is, for example, to calculate the vertical columns using pixel-specific AMFs, including the dependence on observation geometry, surface reflectance, clouds and a priori NO$_2$ vertical profile."

Detailed comments:

Title: The title is misleading, as it suggests that this study is about quantifying ship emissions. Please consider modifying this, especially in the manuscript, for instance in line 149: I would expect to read a number at the end of the section with such a headline.

Thank you for pointing this out. We agree with you. The title of our manuscript has been updated to: "Improved detection of global $NO_2$ signals from shipping in Sentinel-5P TROPOMI data"

The headline of Section 3 has been changed to:
"3 Global shipping signals in filtered TROPOMI $NO_2$ tSCDs"

A statement like that in line 382 should already be made in the abstract.

"qualitatively" has been added to the sentence in Line 5 of the Abstract.

Line 4, line 49: This TROPOMI pixel size only holds for nadir. Towards the swath edges, pixels are considerably larger.

Thank you for this clarification. We added "at nadir" to the sentences in Lines 4 and 49.

Line 90: I agree that the tSCD allows the detection of ship NOx without bias from a-priori knowledge used for the AKs in the operational product. However, as soon as NOx emissions should be quantified, the tSCDs need to be converted to tVCDs by an (appropriate) AMF.

Thank you very much for pointing that out. You are right that an appropriate and pixel-specific AMF is needed to convert the tSCDs into the tVCDs for the quantification of ship emissions. In this study, we only want to qualitatively detect the shipping signals in the TROPOMI data. Therefore, we used the tSCDs to avoid any influences of model-based parameters on the measurements obtained from the operational TROPOMI product. We feel that it is important to first establish that the signals in the satellite data are genuine and then use a priori data (not necessarily from models) where needed.

Please clarify.

Lines 93-104: I think a short modification sentence would be helpful: The tSCD is provided in the operational product (or could easily be calculated by subtracting V_strat*AMF_strat from total SCD), but this quantity turned out to be affected by a simplification in the operational processor: ...

Thank you for this hint. We implemented your suggestion to the sentence in Line 93:
"The tSCD is provided in the operational product, but this quantity turned out to be affected by a simplification in the operational processor, as distinct box-like patterns were noticed in the TROPOMI $NO_2$ data during data processing."

Line 106: What does "approximately" mean?

We use 33 pixels with a resolution of 0.03° for the standard high-pass filter, so the resulting box size is 0.99° (approximately 1°). We thought it might be more helpful for readers to know the degree value rather than the number of pixels used, as it is easier to visualize and more consistent. These numbers (33 pixels for the 1° box size and 9 pixels for the 0.25° box size) are also mentioned in Appendix A1, which explains the high-pass filtering in more detail.

Line 109: How far is this a "saturation"? What is saturated?

The "saturation" of the high-pass filtered tSCDs refers to the saturation of the color scale by values up to 1e14 molec $cm^{-2}$ large. The occurrence of these very high values depends on the region. An example of only high-pass filtered data is shown in Fig. B2. The values greater than 3e13 molec $cm^{-2}$ corresponding to this saturation occur mainly in the south and southeast of the coastline in the Arabian Sea and the Gulf of Oman. We have changed the sentence in Line 109 to:
"However, due to taking the rolling mean on the data, the high-pass filtered tSCDs show a saturation of the color scale by very high $NO_2$ values at some coastlines and negative values around the highlighted shipping lanes."

Filtering: The iterative filter approach seems complex, and Fig. 1 does not really help. I would propose to have a sketch showing the effect of the individual steps on real data instead.

Thank you for this suggestion. We have added illustrations of the various variables resulting from the different filtering steps to the variable names in Figure 1 and updated its caption.

Fig. 2 shows the result of the iterative filter process. I do not understand how the values here can exceed 4e13, when in a previous step everything above 3e13 has been skipped. Please clarify.

The threshold value of 3e13 was only used in Step 2 to mask the highest and lowest values of the high-pass filtered data that localize the shipping routes. After applying this mask to the gridded data in Step 3, the gridded values are interpolated again in Step 4. Therefore, after the following high-pass filtering step with these interpolated values, corresponding values higher than 3e13 reappear in Step 5. This iterative method highlights the shipping lanes more accurately with each repetition. We hope these processing steps are now easier to understand by adding the real data illustrations to Figure 1, as you suggested above.

Line 149: Even with stating the disadvantages of applying filters, one might think that all the discussed filters are applied below. A statement should be added that, as default, they are *not*, with reference to later discussion why not.

We explained in Line 152: "As the standard criterion, no data flagging is applied to the filtered TROPOMI NO$_2$ tSCDs, as discussed later.". However, we have added the following sentence in Line 149 to clarify this beforehand:
"Sect. 3.1 discusses the individual impacts of the various flagging criteria in detail, while no flagging criterion is applied to the tSCDs as a standard in this study."

Line 157: except the model used for assimilating strat. NO2...

That's right! We added "tropospheric" to the sentence.

Fig. 3: The map of resulting ship tracks is quite impressive, and the benefit of 1° over 0.25° quite prominent. However, this is a result of the high-pass filter, and this will inevitably cause *negative* tSCDs next to the shipping lines. This would have to be taken into account for quantification of emissions. I thus think that the figures should also show the negative values to create awareness for this effect - this might not look as nice, but more honest.

You have made a point. Although the negative values are already represented in the cross-sections, we agree that more awareness should be created for them. Therefore, we added a figure to the Appendix, which shows a global map with a symmetrical color scale. In addition, we added the following paragraph in Line 149:
"Another essential aspect to consider is that the choice of the maps' color scale significantly affects the visualization of the shipping lanes. When a symmetric color range with equally distributed negative and positive limits is used, the negative values in some coastal regions become more prominent on the map (see Fig. B1). In contrast, shifting the values to a positive color range highlights the shipping lanes more effectively (see Fig. 2). This study focuses on the qualitative detection of shipping signals. Therefore, a positive color scale is used for the maps."
Furthermore, the following sentence has been added to the caption of Figure 2:
"Fig. B1 shows the corresponding global map of the filtered TROPOMI NO$_2$ tSCDs using a symmetric color scale to visualize also the negative values resulting from the high-pass filtering method, which occur mainly at coastlines around Europe and Asia."
The reference to this Appendix figure has been added to Line 269, where the negative values are also explained in the cross-sections.

CAMS comparison:

Are the CAMS tVCDs high-pass filtered as well? If not, they should for a more meaningful comparison.

Good point; thank you very much for this hint! In the preprint document, only an offset was removed from the CAMS data. We now high-pass filtered the CAMS model data with the same high-pass filtering settings, adding this information in Line 386:
"To ensure a consistent analysis, the CAMS model data are high-pass filtered with the same settings as the TROPOMI data with a tenfold iteration. Consequently, 3 pixels of the CAMS model data are used for the rolling mean applied to the high-pass filter to maintain the 1° box size."
We have changed Figures 11, C1, C2, and C3 and their captions and updated Section 4, the Abstract (Lines 15-18), and the Conclusions section (Lines 468-480) with the modified results and the term "high-pass filtered CAMS data" or the like. In addition, we added a second figure for the South Atlantic shipping lane (Fig. C4), using a decreased threshold for the masking in the high-pass filtering method since we found a large dependence of the CAMS model data on this defined threshold value in this region. We added the following paragraph to Line 418 to discuss this issue:
"For the shipping route in the South Atlantic (Fig. C3c), the peak of the CAMS curve is twice as large as those of the TROPOMI curves, which do not exceed 0.5e13 molec cm$^{-2}$. It appears to be slightly shifted due to the lower number of values resulting from the coarser spatial resolution. The difference between the two data sets is small with these standard settings. However, Fig. C4 shows that the high-pass filtered CAMS NO$_2$ data for this shipping lane strongly depends on the masking threshold used in Step 2 of the high-pass filtering method (see Sect. 2). When the masking threshold value is lowered to ±1e13 molec cm$^{-2}$, the high-pass filtered CAMS NO$_2$ tVCDs increase extraordinarily and are at least three times higher than with the standard threshold of ±3e13 molec cm$^{-2}$ (Fig. C3). The higher peak is accompanied by a much broader FWHM, so the shipping lane expands over the entire polygon area. This large dependence on the choice of masking threshold is only found for the shipping lane in the South Atlantic and only for the CAMS NO$_2$ data when considering the three selected regions. In contrast, the TROPOMI data show slightly higher peaks, where the difference is consistent and comparable to the other shipping lanes. Therefore, it should be kept in mind that applying the high-pass filter to the CAMS model data may result in large changes of the values in some regions, depending on the threshold value chosen."

Line 421: and avoiding artificial ship tracks just introduced by the a-priori profiles?

Thank you. Your suggestion has been added to the sentence:
"By focusing on NO$_2$ tSCD rather than tVCD, a more objective identification of NOx emissions from shipping is obtained, benefiting from the enhanced spatial resolution and avoiding artificial ship tracks introduced by the a priori profiles, associated with limitations of the AMFs derived from the coarse TM5 model, which only inadequately captures localized emissions."

Fig. C1: This comparison is not appropriate, since the TROPOMI data has been high-pass filtered, but the CAMS data not. At least this aspect needs to be clearly discussed.

Thank you for your comment. We have already discussed this point and the changes applied above. All CAMS model data are now also high-pass filtered.

A statement on data availability is missing. I would encourage the authors to make their results available on a data repository so that future studies could reference this dataset with a doi.

The data set has been submitted to the PANGAEA repository, has been reviewed and can be published as soon as the paper manuscript is published. The DOI of the data set has already been implemented in the manuscript but has not yet been registered. The data set can be accessed via the following temporary link: https://www.pangaea.de/tok/e6a3d2eb22f88dc8930d73b32e3775384d6c3642. The paragraph on data availability has been updated to:
"The data sets of the global filtered TROPOMI $NO_2$ tSCDs for different box sizes of the high-pass filter (1°, 0.5°, 0.25°) and for the standard box size of 1° with the various flagging criteria, as displayed for example in Fig. 2, are freely available from PANGAEA (Felden et al., 2023) under the CC-BY-SA license (Latsch et al. (2025), https://doi.org/10.1594/PANGAEA.982514)."

**Other changes made to the manuscript during the review process:**

All figures with maps (Figures 2-7, 9-11, B1-B2, and C1-C3) have been changed: The inland water pixels are now also flagged as NaN values to focus on the ocean regions and to reduce the scatter over the continents.

Additionally, Figure B1 and Figure C1: The surface classification mask of the TROPOMI data is now also applied to the CAMS data. Thus, the same pixels are flagged as NaNs for visual consistency. The reason why these water pixels are marked as invalid is that the TROPOMI surface classification mask defines snow and ice pixels over water as land pixels. In this study, only water pixels are included. However, these flagged pixels are irrelevant for this study because they represent the snow and ice edge and, therefore, are not affected by ship emissions.

We added the following sentence to the Acknowledgments:
"We thank the two anonymous reviewers for their comments and suggestions which helped to improve the results and their presentation in this manuscript."

---

## Author Response (AR2)

**EGUSPHERE-2025-107 Author's comment to Editor decision (Miriam Latsch et al.)**

Legend: Editor comments in **black**, author comments in **blue**, changes in the manuscript text in **green**

Thank you for your revised manuscript and Author's response for the article entitled "Improved detection of global NO2 signals from shipping in Sentinel-5P TROPOMI data". I have reviewed them and am pleased to recommend publishing it with minor technical corrections as follows:

We would like to thank the editor for accepting our manuscript. We have implemented the corrections as follows.

- Page 3, line 69: Please check the citations.

The citations of the references have been corrected.

- Page 8, line 192: "is minimal (December to April)" => "is minimal in summer (December to April)" may be helpful for clarity.

Thank you for this suggestion. We added "in summer" to the sentence.

- Page 22, line 523: "The reasons ... is not yet explained" => "The reasons ... are not yet explained"

Thank you. "is" has been changed to "are".

**Other changes made to the manuscript:**

Line 187: "data indicates" has been changed to "data indicate".